# Necroptosis enhances 'don't eat me' signal and induces macrophage extracellular traps to promote pancreatic cancer liver metastasis

Cheng-Yu Liao [1,2,3,7], Ge Li[4,7], Feng-Ping Kang[1,7], Cai-Feng Lin[1,2,3], Cheng-Ke Xie[1,2], Yong-Ding Wu[1,2], Jian-Fei Hu[1,2], Hong-Yi Lin[1,2], Shun-Cang Zhu[1,2], Xiao-Xiao Huang[1,2,3], Jian-Lin Lai[1,2,3], Li-Qun Chen[3], Yi Huang[1,3], Qiao-Wei Li[1,5,6], Long Huang[1,2,3], Zu-Wei Wang [1,2,3] ✉, Yi-Feng Tian [1,2,3] ✉ & Shi Chen [1,2,3,5,6] ✉

Pancreatic ductal adenocarcinoma (PDAC) is a devastating cancer with dismal prognosis due to distant metastasis, even in the early stage. Using RNA sequencing and multiplex immunofluorescence, here we find elevated expression of mixed lineage kinase domain-like pseudo-kinase (MLKL) and enhanced necroptosis pathway in PDAC from early liver metastasis T-stage (T1M1) patients comparing with non-metastatic (T1M0) patients. Mechanistically, MLKL-driven necroptosis recruits macrophages, enhances the tumor CD47 'don't eat me' signal, and induces macrophage extracellular traps (MET) formation for CXCL8 activation. CXCL8 further initiates epithelial–mesenchymal transition (EMT) and upregulates ICAM-1 expression to promote endothelial adhesion. METs also degrades extracellular matrix, that eventually supports PDAC liver metastasis. Meanwhile, targeting necroptosis and CD47 reduces liver metastasis in vivo. Our study thus reveals that necroptosis facilitates PDAC metastasis by evading immune surveillance, and also suggest that CD47 blockade, combined with MLKL inhibitor GW806742X, may be a promising neoadjuvant immunotherapy for overcoming the T1M1 dilemma and reviving the opportunity for radical surgery.

Pancreatic ductal adenocarcinoma (PDAC) is a devastating form of cancer that is often diagnosed too late, making it difficult to treat effectively, which leads to a poor prognosis[1]. One of the main challenges associated with PDAC is the high incidence of metastasis, especially to the liver, which occurs at the early T stage of the disease (T1M1) and affects nearly 10% of patients[2]. Unfortunately, patients with early liver metastasis are not eligible for radical resection and have very low survival rates. Despite ongoing research, the underlying mechanisms responsible for the early liver metastasis of PDAC remain poorly understood[3].

[1]Shengli Clinical Medical College of Fujian Medical University, 350001 Fuzhou, China. [2]Department of Hepatobiliary Pancreatic Surgery, Fuzhou University Affiliated Provincial Hospital, Fujian Provincial Hospital, 350001 Fuzhou, China. [3]Fuzhou University, 350001 Fuzhou, China. [4]Department of Hepatobiliary Surgery and Fujian Institute of Hepatobiliary Surgery, Fujian Medical University Union Hospital, 350001 Fuzhou, China. [5]Fujian Provincial Center for Geriatrics, 350001 Fuzhou, China. [6]Fujian Key Laboratory of Geriatrics, 350001 Fuzhou, China. [7]These authors contributed equally: Cheng-Yu Liao, Ge Li, Feng-Ping Kang. ✉e-mail: drzuwei123@163.com; tianyifeng@fjmu.edu.cn; wawljwalj@163.com

Early-stage tumour cells are subjected to stressors from anti-tumour immune responses, and it is believed that resistance to cell death is necessary for the acquisition of a metastatic phenotype[4]. We hypothesize that liver metastasis accompanied by a small primary PDAC tumour size may involve increased resistance to cell death, possibly through necroptosis. Necroptosis is a regulated type of pro-grammed cell death during which mixed lineage kinase domain-like pseudo-kinase (MLKL) undergoes phosphorylation and oligomerization, resulting in the formation of cytosolic pores and the uncontrolled release of intracellular substances[5-7]. The activation of necroptosis can theoretically slow tumour progression[8-11], but it may have pro- or anti-tumour effects depending on the tumour and its microenvironment after MLKL activation[12,13]. The role of MLKL in PDAC has not been well characterized.

Intracellular material whose release is triggered by necroptosis shapes the TME, attracting phagocytes, mainly tumour-associated macrophages (TAMs), which bridge innate and adaptive immunity[14]. M1 and M2 TAMs have distinct effects on tumours. Despite the anti-tumour effect of M1 TAMs, tumour cells adapt within the complex TME and are influenced by the balance between pro-tumorigenic and anti-tumorigenic factors[14]. The precise roles of macrophage phenotype in necroptosis-driven TME events are still unclear[13]. Furthermore, tumour cells employ 'don't eat me' signals, such as signals mediated via the CD47-SIRPα axis or CD24-Siglec10 axis, that block antigen presentation by TAMs[15]. TAMs can also generate macrophage extracellular traps (MET), which resemble neutrophil extracellular traps (NET) and are composed of an intricate network of fibrous chromatin backbone structures. Although reports have shown that NETs can awaken dormant tumour cells and foster tumour metastasis[16], our knowledge of METs in the context of tumours is exceptionally limited, as is our understanding of the interplay between necroptosis and METs[17].

In this study, we analyse multidimensional data from patients with early T-stage PDAC with liver metastasis, and reveal the pro-metastatic effect of necroptosis, which is driven by aberrantly elevated MLKL expression. This process enhances the tumour CD47 'don't eat me' signal and induces MET formation, which exerts a crucial role in multiple metastatic stages. Targeting necroptosis alongside CD47 prevents PDAC liver metastasis in preclinical models. These findings elucidate a new macrophages-enabled evading manner under necroptosis and provide a promising neoadjuvant strategy for early T-stage PDAC with liver metastasis that addresses a pressing clinical challenge.

## Results

### Aberrant MLKL-mediated necroptosis is associated with liver metastasis in early T-stage PDAC

To identify genes that may be responsible for liver metastasis in early T-stage PDAC, we performed RNA-seq using clinical PDAC tissue samples from stage T1M1 (primary tumour size ≤ 2 cm with liver metastases, $n = 6$) and stage T1M0 (primary tumour size ≤ 2 cm without liver metastases, $n = 6$) tumours and matched para-cancerous tissues (Fig. 1a−c, Supplementary Table 1). This analysis revealed 2142 differentially expressed genes (DEGs) that were found by KEGG and GO analyses to be associated with necroptosis (Fig. 1d, e, Supplementary Fig. 1a), with MLKL emerging as the most significantly upregulated gene (Supplementary Fig. 1b). We then cross-referenced these 2142 DEGs with genes that were upregulated in PDAC tissues versus normal tissues in both T1M1 and T1M0 PDAC cases, which resulted in a set of 216 overlapping DEGs (Fig. 1f). Once again, these 216 DEGs were enriched in pathways related to necroptosis according to KEGG and GO analyses (Fig. 1g, Supplementary Fig. 1c), and MLKL remained the most significantly upregulated gene among this subset (Fig. 1f).

We investigated the potential mechanisms for MLKL elevation in T1M1-PDAC. The elevated MLKL levels did not appear to be directly linked to the primary driving gene mutation in PDAC, as the proportions of KRAS/TP53 mutations were similar in CCLE- PDAC cell lines (Supplementary Fig. 1d, e) and in T1M0 and T1M1 PDAC cases (KRAS mutation: 4/6 T1M0 PDAC cases vs. 3/6 T1M1 PDAC cases; TP53 mutation: 5/6 T1M0 PDAC cases vs. 5/6 T1M1 PDAC cases; Supplementary Table 1). Also, HEK293T cells that were transfected with wild-type/mutated KRAS (G12C/G12D/G12V) did not reveal significant changes on MLKL expression level (Supplementary Fig. 1f). Next, we eliminated the notion that elevation of MLKL was in response to the TME using an in vivo orthotopic model with immunodeficient mice and immunocompetent mice (Supplementary Fig. 1g). We also investigated the expression of potential transcription factors (TFs) for MLKL that predicted by JASPER database, and failed to found the altered TFs for MLKL according to RNA-seq data (Supplementary Fig. 1h, i). Neither were the previously reported regulatory factors (LncRNA-FA2H-2, LncRNA HABON, CRTC2) for MLKL (Supplementary Fig. 1i)[18-20]. Notably, the similar pre-mRNA levels in T1M0-PDAC and T1M1-PDAC indicated the possibility of an abnormal alternative splicing process on MLKL, which may result in the overexpression of MLKL in T1M1-PDAC (Supplementary Fig. 1j)[21].

We further investigated the role of MLKL, the functional executioner of necroptosis, in the development of liver metastasis in early T-stage PDAC. Our results showed that MLKL expression was significantly upregulated in PDAC tissues compared to normal tissues and pancreatic ductal epithelial cells, and this was consistent with the findings obtained with both a TCGA cohort and our in-house cohort (Supplementary Fig. 2a−d). Elevated MLKL expression correlated with poorer survival, particularly with a tumour length ≤ 2 cm, greater liver metastasis in smaller tumours, lower differentiation, and a more advanced tumour stage (Fig. 1h, Supplementary Table 2, Supplementary Fig. 2e−i). Moreover, we noted an increase in MLKL phosphorylation with increasing MLKL expression, which is indicative of increased necroptosis, in our in-house PDAC cohort (Fig. 1i). Western blotting (WB) analysis of primary tissues from T1M1 and T1M0 tumours confirmed the upregulation of MLKL and its phosphorylated form, p-MLKLS357/S358/T360 (Fig. 1j), suggesting the activation of necroptosis. The levels of other necroptosis-related molecules, including RIPK1, RIPK3, p-RIPK1, and p-RIPK3, did not significantly differ. These findings suggest that an aberrant elevation of MLKL-mediated necroptosis plays a critical role in patients with T1M1 PDAC.

To investigate the role of MLKL-mediated necroptosis in T1M1 PDAC, we created organoids from the corresponding clinical primary tumour tissues of patients with T1M1 and T1M0 PDAC that were used for RNA-seq (Fig. 1k). With the elevated MLKL expression level, T1M1 PDAC patient-derived organoids (PDOs) with activated necroptosis exhibited significantly reduced growth rates (Fig. 1k, Fig. 2a). Treatment with GW806742X (GW), an MLKL inhibitor, effectively reversed this arrest in growth and increase in the necroptosis level (p-MLKL level) in T1M1 PDOs, while treatment with a RIPK1/3 inhibitor (Nec-1, GSK-872) did not (Fig. 2a). These results suggest that an aberrant increase in MLKL suppresses tumours by promoting necroptosis and that MLKL activation relies on an RIPK3-independent mechanism in T1M1 PDAC.

To further explore this hypothesis, we first investigated RIPK1, RIPK3 and MLKL expression in TCGA PDAC samples and CCLE PDAC cell lines, and then, we used the RIPK3null PANC1 cell line and the RIPK3high AsPC-1 cell line and simultaneously ectopically overexpressed MLKL to mimic the aberrant elevation of MLKL-driven necroptosis in T1M1 PDAC (Supplementary Fig. 3a−c). MLKL overexpression did not affect RIPK1/3 expression or phosphorylation but did increase the level of p-MLKL level and MLKL-oligomers (Fig. 2b). MLKL overexpression did not significantly alter the apoptosis level, based on determination of the levels of cleaved caspase-3 and cleaved caspase-8 (Fig. 2b). MLKL overexpression induced a significant increase in necroptosis that was characterized by plasma membrane perforation, cytosolic leakage, cell swelling, and nuclear rupture (Fig. 2c); these effects were accompanied

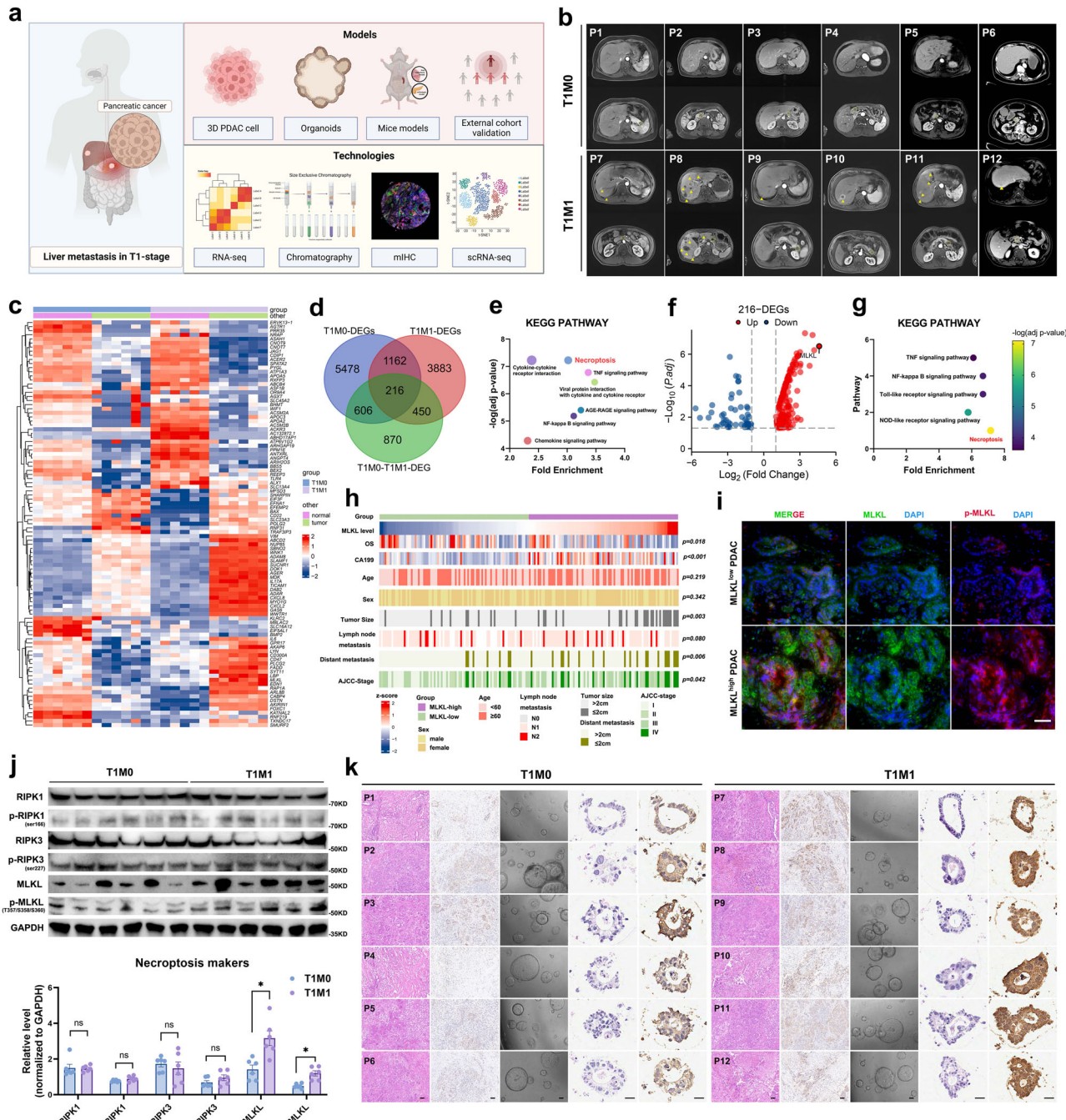

**Fig. 1 | Aberrant MLKL elevation-mediated necroptosis is associated with liver metastasis of early T stage PDAC. a** The overall experimental scheme used in this study. **b** MRI of T1M1 (*n* = 6) and T1M0 (*n* = 6) PDAC patients. Yellow arrow: liver metastasis; yellow circle: primary pancreatic tumour. **c** Matrix diagram of RNA-seq data from T1M0-PDAC (*n* = 6) and T1M1-PDAC (*n* = 6) samples. **d** Venn diagram illustrating the overlap between differentially expressed genes (DEGs) between the tumour and normal groups and the T1M1 and T1M0 PDAC groups. **e** KEGG pathway enrichment analysis of DEGs between the T1M0 and T1M1 PDAC groups. **f** Volcano plot of the final 216 DEGs. **g** KEGG pathway enrichment analysis of the final 216 DEGs. **h** Heatmap of clinicopathological features, genetic profiles, and survival information among groups with different MLKL expression levels (*n* = 170) in the cohort from our centre. The two-sided Pearson chi-square test and were used to assess the differences between the two groups; overall survival (OS) probability was compared using log-rank test, two-sided. **i** Representative images of immuno-

stained MLKL and p-MLKL(T357/S358/S360) in the PDCA cohort (*n* = 20) from our centre; Scale bar: 50 µm. **j** WB and quantification of necroptosis markers in tumour samples from patients with T1M0-PDAC (*n* = 6) and T1M1-PDAC (*n* = 6).
**k** Representative images following H&E staining and IHC staining (MLKL) of T1M0 and T1M1 PDAC primary tumour organoids. Organoids were constructed from primary tumours that performed RNA-seq of T1M1 and T1M1 PDAC patients who underwent MRI. *n* = 6 biologically independent samples for each group, Scale bar: 50 µm. Unless specified otherwise, the data are presented as means ± SEM (error bar) and compared using the two-sided Student's *t* test; *P < 0.05; **P < 0.01; and ***P < 0.001; ns, no significance. **a** Created with BioRender.com released under a Creative Commons Attribution-NonCommercial-NoDerivs 4.0 International license (https://creativecommons.org/licenses/by-nc-nd/4.0/deed.en). Source data are provided as a Source Data file.

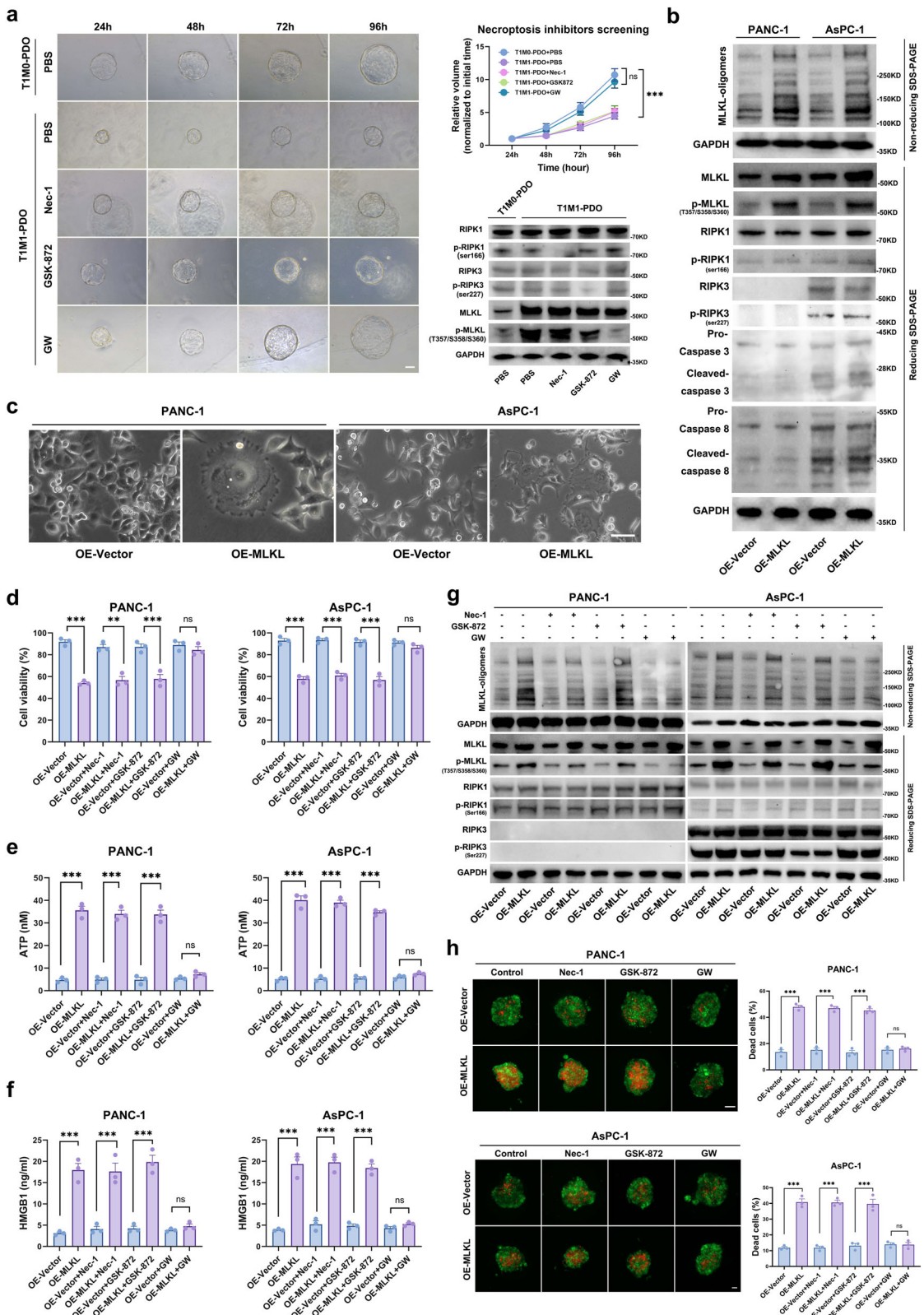

by increased MLKL phosphorylation, MLKL oligomer levels, necrosis, and ATP and HMGB1 release and reduced cell viability, which were rescued by GW but not by Nec-1 or GSK-872 (Fig. 2d–h). Furthermore, we did not observe a similar MLKL expression pattern and RIPK3-independent necrosis mode in pancreatic ductal epithelial cells (HPNE cell line), which further supported the specificity with which an

aberrant elevation in MLKL mediates necroptosis in tumour cells rather than in pancreatic ductal epithelial cells (Supplementary Fig. 2c, Supplementary Fig. 3d). Taken together, these results suggest that necroptosis mediated by an aberrant elevation of MLKL (MLKL-driven necroptosis) in tumour cells is associated with liver metastasis in early T-stage PDAC.

**Fig. 2 | Necroptosis mediated by an aberrant elevation of MLKL is RIPK3-independent. a** Representative images of T1M1 PDAC PDOs and T1M0 PDAC PDOs at 24 h and 96 h after treatment with PBS (control) or different necroptosis inhibitors. The inhibitors Nec-1 (50 μM, a RIPK1 inhibitor), GSK-872 (3 μM, a RIPK3 inhibitor), and GW (1 μM, an MLKL inhibitor) were applied, and a diagram shows construction of the PDOs used for screening of the necroptosis inhibitors. Changes in the relative volume of T1M1 PDAC PDOs and T1M0 PDAC PDOs that received different treatments and the results of WB analysis of necroptosis markers in PDOs treated with necroptosis inhibitors are shown; $n = 6$ biologically independent samples for each group, Scale bar: 50 μm. **b** WB analysis of necroptosis and apoptosis markers; $n = 3$ biologically independent samples. **c** Cell morphology. Scale bar: 25 μm. **d** Cell viability assay at 72 h after MLKL transfection; $n = 3$ biologically independent samples. **e** ATP levels in the supernatant 72 h after MLKL transfection; $n = 3$ biologically independent samples. **f** HMGB1 levels in the supernatant 72 h after MLKL transfection, $n = 3$ biologically independent samples. **g** WB analysis of necroptosis markers after 48 h treatment with different necroptosis inhibitors. The inhibitors used were Nec-1 (50 μM, a RIPK1 inhibitor), GSK-872 (3 μM, a RIPK3 inhibitor), and GW (1 μM, an MLKL inhibitor); $n = 3$ biologically independent samples. **h** The necrosis of 3D tumour spheroids based on propidium iodide staining (red) and quantification of the data; $n = 3$ biologically independent samples; Scale bar: 100 μm. Unless specified otherwise, the data are presented as means ± SEM (error bar) and compared using the two-sided Student's $t$ test; $*P < 0.05$; $**P < 0.01$; and $***P < 0.001$; ns, no significance. Source data are provided as a Source Data file.

## Immunocompetence is indispensable for the ability of MLKL-driven necroptosis to promote PDAC metastasis

Because MLKL is associated with increased liver metastasis but decreased tumour size (Supplementary Table 2), we further investigated the impact of MLKL-driven necroptosis on the proliferation and metastatic capacity of PDAC cells. T1M1 PDAC tissue-derived tumour cells formed smaller aggregates but had a more invasive phenotype than did T1M0 PDAC-derived tumour cells in an organoid homologous Matrigel (Fig. 3a). In vitro, MLKL overexpression decreased cancer cell growth, disrupted intercellular junctions, and induced a mesenchymal spindle-like morphology (Fig. 3b); measurement of proliferating cell nuclear antigen (PCNA) levels and a 3D tumour sphere assay reflected the effects of MLKL overexpression on PDAC: it inhibited growth but enhanced invasiveness (Fig. 3c, d). Conversely, MLKL knockout decreased tumour initiation and metastatic capacity (Supplementary Fig. 4a–f).

We assessed necroptosis in liver metastases using NOD-SCID and C57BL/6 mice as models. PDAC tumour cells gathered from the KRAS$^{G12D/+}$; p53$^{R172H/+}$; pdx-1Cre/+ (KPC) spontaneous cancer mouse model, overexpressing MLKL or control vectors that can be tracked by immunofluorescence, were injected into the spleen. Mice with MLKL overexpression exhibited weaker fluorescence in the spleen, consistent with the in vitro results (Fig. 3e). Notably, C57BL/6 mice with MLKL overexpression exhibited increased liver metastasis, while NOD-SCID mice showed no significant difference in liver metastasis, indicating that the role of MLKL in promoting metastasis is dependent on the immune system (Fig. 3e). Furthermore, examination of the spleen revealed increased macrophages infiltration in mice with MLKL overexpression (Fig. 3f), and it was consistent with previous reports[22]. GW treatment and macrophage depletion reversed the increase in liver metastasis without affecting primary tumour lesions in the spleen (Fig. 3g, Supplementary Fig. 4g). To further investigate the macrophage-dependent pro-metastatic effect of MLKL-driven necroptosis, in vitro co-culture experiments were performed, and the results demonstrated an increase in the invasive behaviour of MLKL-overexpressing PDAC cells when co-cultured with macrophages under 2D or 3D conditions, as indicated by cytoskeleton staining. This effect was reversible with GW treatment (Fig. 3h, Supplementary Fig. 4h).

In our orthotopic xenograft C57BL/6 mouse model, MLKL-driven necroptosis had a significant impact on metastatic capacity (Supplementary Fig. 4i). The MLKL-overexpressing mice exhibited the increased occurrence of massive liver metastasis (Fig. 3i, Supplementary Fig. 4j), and conversely, MLKL knockout mice showed decreased tumour burden of liver metastasis (Supplementary Fig. 4e, f). Regarding the primary tumour size, smaller primary tumours were observed in OE-MLKL group (Fig. 3i), whereas MLKL knockout did not significantly promote primary tumour size in vivo (Supplementary Fig. 4e, f). This may suggest a more intuitive link between MLKL-driven necroptosis and liver metastasis capacity. The increased macrophages infiltration in mice with MLKL overexpression was also observed (Supplementary Fig. 4k). GW treatment and macrophage depletion

reduced the incidence of liver metastasis (Fig. 3j). These results indicate that immunocompetence is indispensable for liver metastasis promoted by MLKL-driven necroptosis, which reflects a mechanism where a form of sacrificial harm promotes metastasis.

## Crosstalk between cancer cells and macrophages induced by MLKL-driven necroptosis promotes macrophage infiltration and activation

To unravel how MLKL-driven necroptosis promotes immune-dependent PDAC liver metastasis, we conducted scRNA-seq of primary tumours from an orthotopic C57BL/6 mouse model ($n = 3$ for each group; Fig. 4a). From orthotopic tumours samples, we identified 115,691 single cells that were clustered into 28 unique clusters (Fig. 4b, c, Supplementary Fig. 5a, b). Then we further identified cancer cells (aneuploid) according to copy number variation (CNV) analysis on those clusters (2, 4, 8, 10, 16, and 17) that exhibited significant enrichment in epithelial gene transcripts, including *EHF, PIL5, CST3, TNC, LPLI, IL11, CADM1, COL18A1, GM6093, FST, GM49504, NEAT1* and *TSC22D3* (Fig. 4d, e, Supplementary Fig. 5c, d). Among the other 22 clusters, which represented various cell types, the significant difference between the OE-Vector group and the OE-MLKL group was observed in the macrophage population (Fig. 4f, Supplementary Fig. 5e, f).

Further analysis of the macrophage population revealed distinct polarization in the OE-Vector and OE-MLKL groups, with more M2 macrophages in the former and more M1 macrophages in the latter (Fig. 4g, h, Supplementary Fig. 5g). We then examined the macrophages by single-cell trajectory analysis (Fig. 4i). According to the pseudotime trajectory of the OE-MLKL group, an increase in the M1 marker gene *H2-Aa* suggested increased M1 polarization (Fig. 4i). Additionally, RNA-seq analysis of T1M1 and T1M0 PDAC patient tumours demonstrated greater M1 macrophage infiltration score in the T1M1 PDAC patient tumours based on QuanTIseq analysis (Fig. 4j).

Next, we co-cultured PANC-1 cells with THP-1-derived macrophages (M0) for 72 h and analysed the macrophages by RNA-seq (Fig. 4k). Co-culture with MLKL-overexpressing PANC-1 cells resulted in upregulation of the M1 macrophage marker CD86 and downregulation of the M2 macrophage marker CD163 (Fig. 4l, Supplementary Fig. 6a, b). KEGG analysis indicated enrichment in macrophage activation, and GO analysis indicated enrichment in various cytokine and chemokine signalling pathways, suggesting an immune response to MLKL-driven necroptosis and macrophage recruitment (Fig. 4m, n). Flow cytometry further validated the increased M1 polarization of macrophages that were co-cultured with PANC-1-OE-MLKL or AsPC-1-OE-MLKL cells (Supplementary Fig. 6c). QuanTIseq analysis of a PDAC TCGA dataset also demonstrated a positive correlation between MLKL expression and M1 macrophage infiltration in PDAC (Supplementary Fig. 6d).

When we assessed the effect of conditioned medium (CM) derived from PDAC cells overexpressing MLKL and observed increased macrophage recruitment capacity (Supplementary Fig. 6e). The PANC-1-OE-MLKL and AsPC-1-OE-MLKL cell lines presented elevated levels of

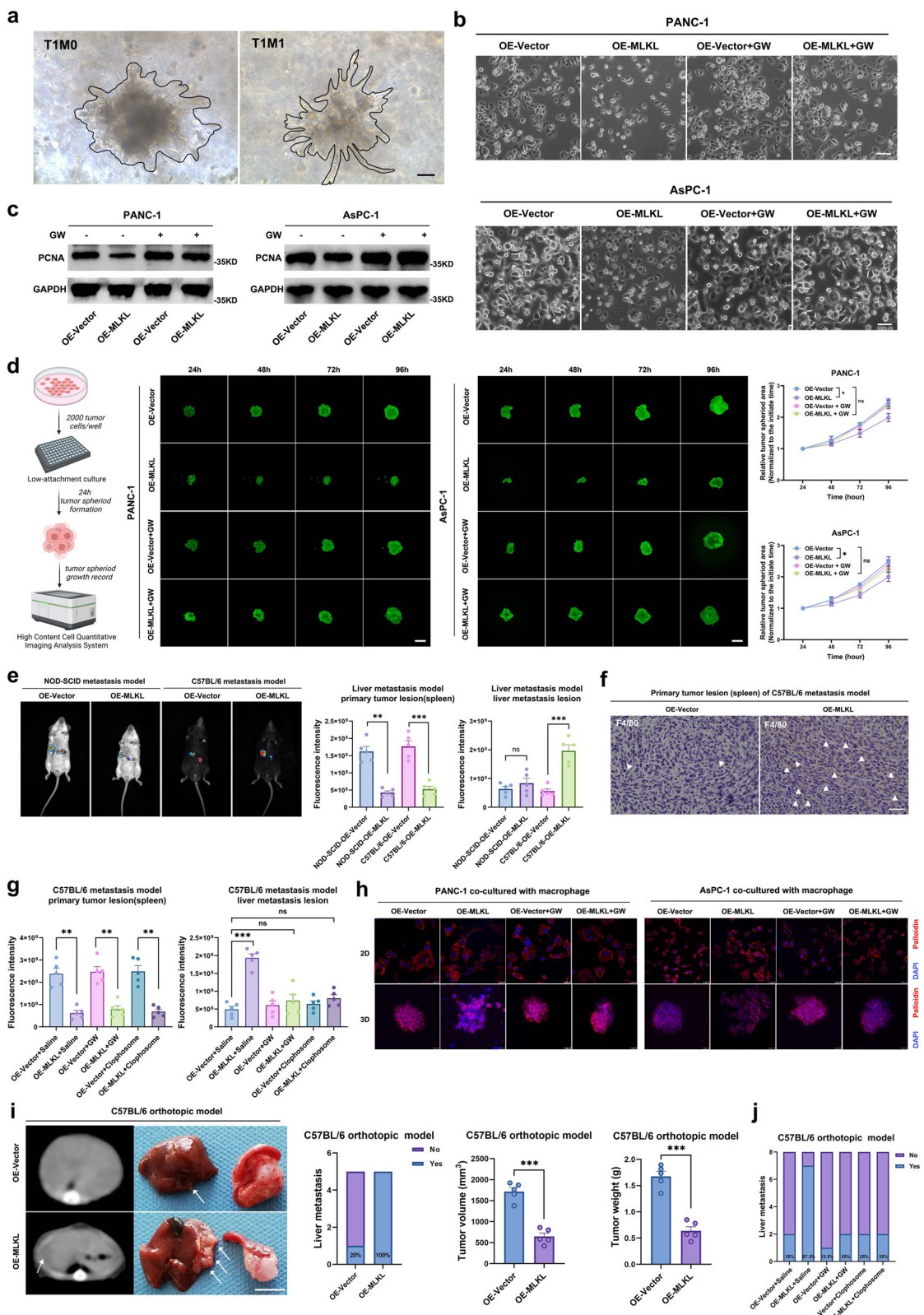

macrophage chemokines (CXCL1, CXCL2, CXCL8, CCL2, and CCL5), and these effects were reversed by GW treatment (Supplementary Fig. 6f, g). Separate KEGG and GO analyses of the macrophages in clusters 5, 9, 11, 19, and 27 revealed significant changes in phagosomes, lysosomes, and processes such as antigen presentation processing and the activation of immune responses (Supplementary Fig. 6h, i), suggesting strong crosstalk between cancer cells and macrophages.

Overall, the results indicate that MLKL-driven necroptosis appears to induce macrophage infiltration and activation.

### MLKL-driven necroptosis promotes phagocytosis resistance by enhancing the tumour 'don't eat me' signal (CD47 upregulation)

MLKL-driven necroptosis in tumour cells leads to the release of the cell contents, cytokines, antigens, and debris, attracting phagocytic cells

**Fig. 3 | MLKL-driven necroptosis leads to immune-dependent promotion of PDAC liver metastasis despite suppression of cell proliferation.**
**a** Representative morphology of tumour cells from T1M1 and T1M0 PDAC seeded in organoids Matrigel. *n* = 6 each group; Scale bar: 25 μm. **b** Cell morphology; *n* = 3 biologically independent samples. Scale bar: 25 μm. **c** WB analysis of PCNA in cells untreated/treated with GW (1 μM) for 48 h; *n* = 3 biologically independent samples. **d** Diagram and the growth of 3D tumour spheroid formation assay. Quantified growth curve was evaluated by the tumour volume normalized to the initial volume; *n* = 3 biologically independent samples; Scale bar: 100 μm.
**e** Bioluminescence imaging and quantification of liver metastasis in representative NOD-SCID and C57B/L6 mice 3 weeks after the injection in liver metastasis model; *n* = 5 mice per group. **f** Representative F4/80 IHC images; Triangles indicate F4/80-positive macrophages; Scale bar: 50 μm; *n* = 5 mice per group. **g** Bioluminescence of liver metastases and primary spleen nodes of mice received different treatments: saline, GW (100 μM in 50 μl, iv. 3×/week), and depleted of macrophages (clodronate liposomes, 200 μl iv.) three weeks after the injection in liver metastasis model; *n* = 5

mice per group. **h** Immunostaining of F-actin (Palloidin, red) and nuclei (DAPI, blue) in 2D tumour cells and 3D tumour spheroids after indirect co-culture with macrophages; *n* = 3 biologically independent samples; Scale bar: 25 μm. **i** Micro-CT, gross morphology, and incidence of liver metastases, primary tumour volume and weight quantification in the orthotopic model; White arrow indicates the liver metastasis; The mice were euthanized at 4 weeks after injection; *n* = 5 mice per group; Scale bar: 5 mm. **j** Incidence of liver metastasis in orthotopic model after different treatments: saline, GW (100 μM in 50 μl, iv. 3×/week), or macrophage depletion (clodronate liposomes, 200 μl, iv.); *n* = 8 mice per group; mice were euthanized at 4 weeks after injection. All data are presented as means ± SEM (error bar) and compared using the two-sided Student's *t* test; *P < 0.05; **P < 0.01; and ***P < 0.001; ns, no significance. **d** Created with BioRender.com released under a Creative Commons Attribution-NonCommercial-NoDerivs 4.0 International license (https://creativecommons.org/licenses/by-nc-nd/4.0/deed.en). Source data are provided as a Source Data file.

for clearance[5]. We explored the interaction between PDAC cells and macrophages, specifically between tumour-killing M1 TAMs. In co-culture models, we observed no further reduction in tumour cell viability or proliferation but did observe an increase in the invasive phenotype, as we previously reported (Fig. 5a, b, Supplementary Fig. 7a). Macrophages, particularly M1 TAMs, can clear debris and kill tumour cells that present antigens[23,24]. Analysis of macrophages from mouse tumours with the control vectors and MLKL overexpression revealed significant differences in antigen presentation pathways through scRNA-seq (Supplementary Fig. 7b, c). We investigated phagocytosis and found that MLKL overexpression promoted tumour cell escape from macrophage phagocytosis, but this effect was abrogated by GW treatment (Fig. 5c–f, Supplementary Fig. 7b, c).

To investigate the mechanism underlying the reduction in phagocytosis, we first measured the phagocytosis of GFP-labelled *E. coli* by macrophages, and the results showed an intact macrophage phagocytosis capability (Supplementary Fig. 7d). This prompted an investigation into the antiphagocytic potential of tumour cells. We focused on the 'don't eat me' signal—the CD47/SIRPα axis and CD24/Siglec10 axis (25). Our RNA-seq results revealed the upregulation of CD47 but no changes in SIRPα, CD24, or Siglec10 expression (Supplementary Fig. 1b, Fig. 4l). In co-culture experiments, CD47 expression was increased in PANC-1-OE-MLKL cells, while the expression of other factors remained unchanged (Fig. 5g). The pseudotime trajectory analysis showed a similar SIRPα expression in the macrophages and an increase in CD47 expression in cancer cells from the OE-MLKL C57BL/6 orthotopic model (Fig. 5h). Additionally, an increase in CD47 expression was observed in co-culture experiments, and this change was reversed by the addition of GW (Fig. 5i). Nevertheless, the use of anti-CD47 rescued the increased resistance to phagocytosis (Fig. 5j).

Interestingly, when tumour cells were cultured alone, CD47 expression remained unchanged, indicating that MLKL alone does not upregulate CD47 in tumour cells (Supplementary Fig. 7e, f). Thus, we further investigated the regulatory effect of CM derived from the co-culture experiments on CD47 (Supplementary Fig. 7e). The CM derived from co-culture experiments consisting of PANC-1-OE-MLKL cells and macrophages upregulated CD47 expression to a greater extent, but this effect was abolished by GW treatment (Fig. 5k). These findings suggested that CM plays a role in upregulating macrophage CD47 expression, potentially through cytokines. Notably, among known CD47-upregulating cytokines (IL-6, IL-1β, TNF and IFN-γ)[25], IL-6 was found to be the most elevated in the CMs derived from co-culture experiments and responsible for the upregulation of CD47 in the context (Fig. 5l, m). Further investigation revealed that IL-6 primarily originated from the co-cultured macrophages and not from necrotic cells (Supplementary Fig. 7g, h). In vivo, CD47 was also found to be upregulated, and this effect was reversed by anti-IL6 and GW treatment (Fig. 5n). Furthermore, liver metastatic lesions maintained

high CD47 expression (Supplementary Fig. 7i). Consistent with tumour tissue WB analysis, the immunofluorescence analysis of PDOs also revealed upregulated CD47 expression in T1M1 PDAC tumours (Fig. 5o, Supplementary Fig. 7j). These findings suggest that while MLKL-driven necroptosis recruits and activates more macrophages, which may contribute to tumour cells developing resistance to phagocytosis and ensuring their survival.

## MLKL-driven necroptosis induces the formation of macrophage extracellular traps (METs) that promote tumour metastasis

Surviving from phagocytosis doesn't necessarily imply metastasis, and we hypothesized that a cancer cell "survival first, then metastasis" might be in play. Our focus shifted to tumour cell changes after acquiring phagocytosis resistance. 3D invasion and migration assays revealed that MLKL-overexpressing tumour cells had greater metastatic capacity when co-cultured with macrophages, and this effect was reversed by GW treatment (Fig. 6a, b). Pseudotime analysis indicated an EMT trajectory, which was supported by in vitro WB analysis (Fig. 6c, d). MLKL-overexpressing tumour cells that co-cultured with macrophage exhibited a higher incidence of liver metastasis in mice, which was absent in GW-treated counterparts (Fig. 6e), suggesting enhanced metastatic ability post-macrophage interaction.

When examining morphological changes, we noticed the formation of reticular structures around macrophages when these macrophages were co-cultured with MLKL-overexpressing tumour cells, resembling macrophage extracellular traps (METs). We confirmed these structures to be METs through FISH staining of MET markers (citrullination of histone H3, CitH3 and MMP2), which colocalized (Fig. 6f). Multiple MET markers (CitH3, MPO, MMP2, MMP9, and MMP12) were elevated in macrophages that were co-cultured with tumour cells, but this effect could be rescued by treatment with MET inhibitor (Cl-amidine, a specific PADI4 inhibitor) or GW (Fig. 6g, h). However, tumour cells did not show an increase in the expression of these markers (Supplementary Fig. 8a, b). Considering the similar consequences (DNA and cytokine/chemokine release) of necroptosis and METs, we also ruled out changes in necroptosis in the macrophages (Supplementary Fig. 8c)[26–33]. Fluorescence staining of primary tumour tissue from MLKL-overexpressing C57BL/6 orthotopic model mice revealed an increase in METs, which was abolished by treatment with MET inhibitor or GW (Supplementary Fig. 8d). SYTOX-Orange staining also indicated an increase in the DNA scaffold, a key component of METs, in macrophages co-cultured with OE-MLKL tumour cells, and this effect was reversed by GW treatment (Fig. 6i). Plasma CitH3 levels in C57BL/6 orthotopic models showed that macrophage depletion, not neutrophil depletion, rendered CitH3 increase, supporting METs formation over NETs (Fig. 6j). These evidence suggested that MLKL-driven necroptosis induces METs formation.

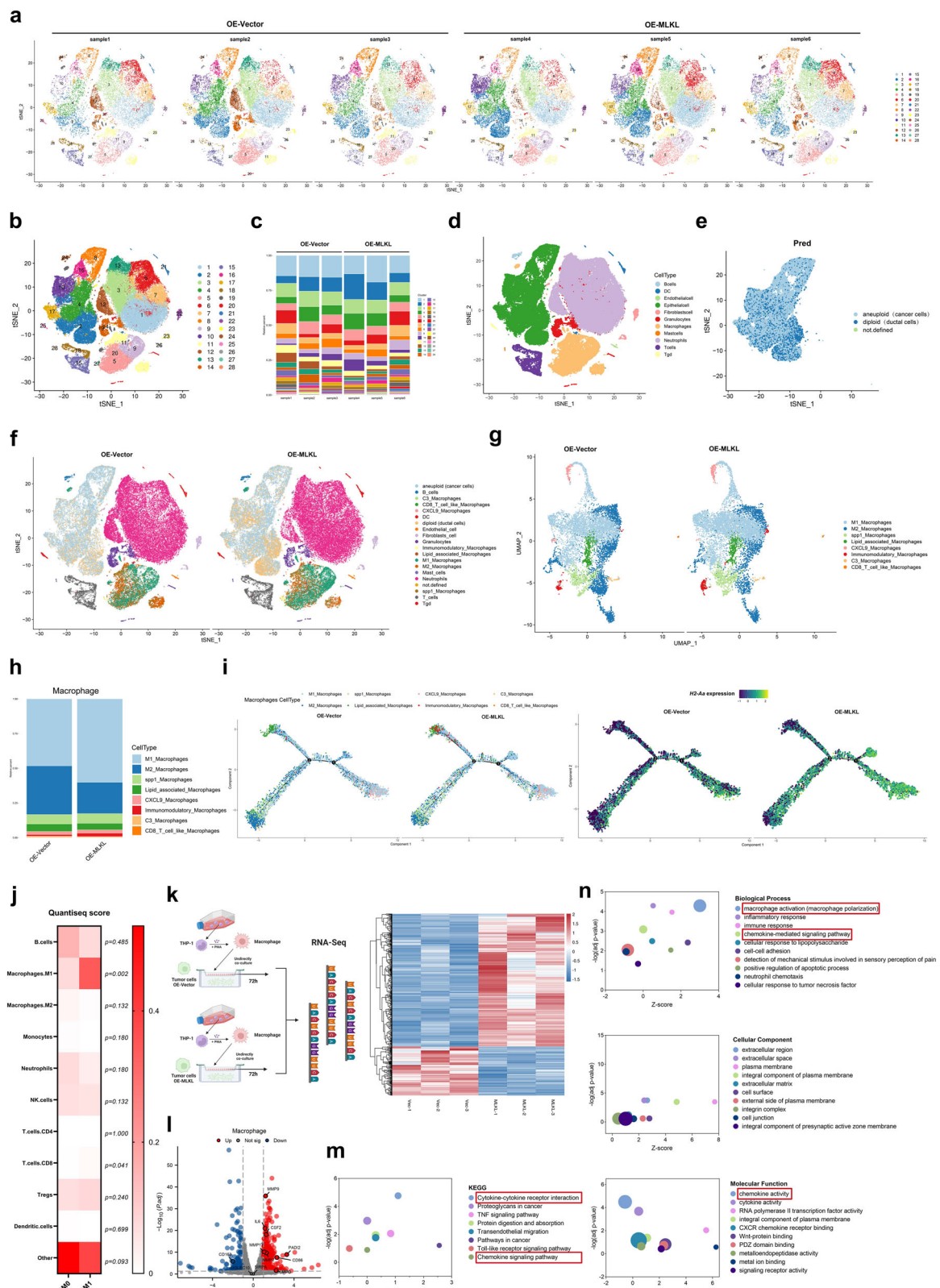

To comprehend METs formation in the PDAC microenvironment, we examined the impact of MLKL-driven necroptosis on macrophages. CM from cancer cells undergoing necroptosis increased METs markers in macrophages (Supplementary Fig. 8e−g), and CM from tumour cells treated with GW reversed this effect (Supplementary Fig. 8h), indicating a paracrine mechanism involving necroptosis-released cytokines in METs induction. We delineated CM components using cytokine antibody arrays, revealing significant increases in CXCL8, CCL5, CCL2, TGF-β1, GRO et al. (Supplementary Fig. 9a). Among these cytokines, CXCL8 is known to promote METs formation in pathogenic infections, while CCL5, CCL2, and GRO recruit macrophages and promote macrophage M1 polarization[27]. We confirmed the increase in the transcription of these genes in MLKL-overexpressing tumour cells (Supplementary Fig. 9b). Neutralizing CXCL8 in CM derived from

**Fig. 4 | MLKL-driven necroptosis promotes crosstalk between cancer cells and macrophages, leading to macrophage infiltration and activation. a** t-SNE plots of tumour samples from the C57BL/6 orthotopic model mice were generated via scRNA-seq ($n = 3$ per group). The mice were euthanized 4 weeks after injection. **b** t-SNE plots of the 28 identified cluster cell populations obtained from the scRNA-seq data. **c** The proportions of the 28 identified clusters in each sample. **d** Eleven cell populations were obtained by manual merging in a t-SNE plot on the basis of corresponding marker gene expression. **e** The epithelial cells were defined into cancer cells (aneuploid) and pancreatic duct cells (diploid) according to the CNV analysis. **f** tSNE plots from scRNA-seq analysis of OE-vector tumours and OE-MLKL PDAC tumours in C57BL/6 orthotopic model mice and the proportions of the cell types. Aneuploid, cancer cells. **g** UMAP plots from scRNA-seq analysis of OE-vector tumours and OE-MLKL PDAC tumours in C57BL/6 orthotopic model mice and the proportions of the cell types. **h** The proportions of different defined macrophage types in the macrophage population. **i** The single-cell trajectory of macrophages

coloured by original cluster identity and the pseudotime trajectory of macrophages. The expression of *H2-Aa* was projected onto single-cell trajectories. Gene expression values are scaled and log-normalized. **j** Tumour-infiltrating immune cell score matrix on the basis of T1M0-PDAC ($n = 6$) and T1M1-PDAC ($n = 6$) RNA-seq data generated via the QuanTIseq method. The mean of QuanTIseq score in the two groups were shown in the heatmap and compared with two-sided Mann–Whitney $U$ test. **k** Workflow of the macrophages used for RNA-seq and heatmap showing the DEGs in macrophages co-cultured with PANC-1-OE-Vector cells and PANC-1-OE-MLKL cells for 72 h, after which RNA-seq was performed ($n = 3$ for each group). **l** Volcano plot of the DEGs. **m** KEGG pathway enrichment analysis of the DEGs. **n** GO enrichment analysis of the DEGs. **k** Created with BioRender.com released under a Creative Commons Attribution-NonCommercial-NoDerivs 4.0 International license (https://creativecommons.org/licenses/by-nc-nd/4.0/deed.en). Source data are provided as a Source Data file.

MLKL-overexpressing tumour cells that were cultured alone failed to induce MET formation, as seen in WB analysis and SYTOX staining (Supplementary Fig. 9c, d). In vivo, treatment with an anti-CXCL15 antibody (CXCL15, a homologue of human CXCL8) abolished the METs formation in primary C57BL/6 orthotopic tumours (Supplementary Fig. 9e). These results suggested that CXCL8 release induced by MLKL-driven necroptosis triggers MET formation.

We examined whether increased METs formation in an M1 TAM-enriched microenvironment enhances the metastatic ability of MLKL-overexpressing tumour cells. The results showed that inhibiting METs with Cl-amidine, a specific PADI4 inhibitor, attenuated MLKL overexpression-induced liver metastasis in vivo and mitigated MLKL overexpression-induced EMT in vitro (Supplementary Fig. 9f, Fig. 6k). This inhibition also reduces liver metastasis in NOD-SCID mice injected with MLKL-overexpressing tumour cells conditioned with METs-containing CM (Fig. 6l), confirming that MLKL-driven necroptosis induces MET formation, enhancing tumour cell metastasis. This was further supported by the similar patterns characterized by increased EMT and METs in T1M1-PDAC tumour tissues and in vivo IHC results from orthotopic models (Supplementary Fig. 9g, h). These findings suggest that MLKL-driven necroptosis promotes METs formation, bolstering PDAC cells' metastatic capability while evading macrophage phagocytosis.

## MLKL-driven necroptosis enhances EMT and endothelial adhesion through CXCL8

We then delved into the consequences of METs formation. We treated wild-type PANC-1 cells with METs-containing CM and performed RNA-seq and WB analysis (Fig. 7a, b). RNA-seq analysis revealed upregulated metastasis-related terms in KEGG and GO databases, such as "positive regulation of EMT", "cell adhesion", "cell migration", "ECM organization", and "collagen degradation", which led us to hypothesize that METs-containing CM can facilitate metastasis at various stages (Fig. 7c, d). Among the DEGs, the expression of EMT markers, including *N-cadherin, VIM, Snail*, and *ZEB1*, was upregulated, which was confirmed by WB analysis (Fig. 7e, f). The expression of these EMT markers remained unchanged in MLKL overexpressing PDAC cells when cultured alone (Fig. 7g). These findings suggested that MLKL alone is not sufficient to induce PDAC cell EMT and it was the series of cytokines induced by necroptosis microenvironment that conveys the EMT signal to nearby surviving tumour cells, thereby supporting metastasis.

We subsequently investigated how METs formation affected the cytokines and chemokines components in TME. Cytokines and chemokines in CM from co-cultured cells exhibited elevated levels of inflammation-related cytokines (IL-6, CXCL8, GM-CSF, CCL-8, CCL-7, and CCL-2) (Fig. 7h), which are known to enhance EMT through their receptors[34–36]. To understand METs' role in promoting metastasis, we used MET inhibitor (Cl-amidine) as a perturbation approach and identified the elevation of CXCL8, IL-6 and GM-CSF were significantly contributed by METs (Fig. 7i, Supplementary Fig. 10a). These cytokines

(CXCL8, IL-6, and GM-CSF) are well documented to promote tumour metastasis by enhancing tumour cell EMT through their receptors (CXCR1/2, IL6R, and GM-CSFR, respectively)[37–45]. Interestingly, receptor expression on MLKL-overexpressing tumour cells remained unchanged (Supplementary Fig. 10b). Individually neutralizing CXCL8, GM-CSF, or IL-6 did not completely inhibit the increase in EMT promoted by co-culture-derived CM (Fig. 7j, Supplementary Fig. 10c). While the MET inhibitor inhibited EMT to some extent, it was not as potent as GW. This suggests that MLKL-driven necroptosis enhances PDAC's metastatic capability by orchestrating various cytokines' effects, including those associated with METs.

Next, we further sought to identify the key cytokine-receptor pathway that is responsible for driving metastasis beyond EMT regulation. By analysing RNA-seq data of T1M1-PDAC and T1M0-PDAC, our KEGG analysis identified the CXCR pathway as the sole cytokine-related pathway (Supplementary Fig. 10d), which was reported to be the chemokine receptors of CXCL family that enhances tumour metastasis in muti-steps including EMT, intracellular adhesion, angiogenesis to support distant metastasis[46–48]. Among the cytokines and chemokines whose abundance was significantly increased, CXCL8 is a widely recognized ligand for CXCR1/2 receptors and has dual pro-tumorigenic effects, including promoting tumour angiogenesis and facilitating tumour cell adhesion across the endothelium through CXCR1/2 receptors[47,49]. Hence, we hypothesized that the CXCL8-CXCR1/2 axis plays an important role in liver metastasis.

Cell adhesion changes were evident in GO analysis (Fig. 7c) and CXCL8 has been reported to upregulate ICAM1, facilitating adhesion between PDAC cells and endothelial cells, leading to PDAC liver metastasis[50,51]. According to our RNA-seq data, in addition to CDH1/CDH2 (E-cadherin/N-cadherin), ICAM1 was upregulated in both T1M1 PDAC tissues and MET-containing CM-treated PDAC cells, which indicated that METs may affect tumour cell adhesion through ICAM1 (Fig. 7e, Fig. 7k, Supplementary Fig. 10d). We found that CXCL8 increased ICAM1 expression in MLKL-overexpressing tumour cells that were co-cultured with macrophages, while no changes were observed when MLKL-overexpressing tumour cells were cultured alone (Supplementary Fig. 10e). This implies that CM from the co-culture experiments could upregulate ICAM1 in tumour cells. The treatment of wild-type PANC-1 and AsPC-1 cells with CM from the co-culture experiments increased ICAM1 expression on these cells (Supplementary Fig. 10f). Neutralizing CXCL8 in the CM or applying CM from co-culture experiments with MET inhibitor or GW treatment abolished the increase in ICAM1 (Supplementary Fig. 10f). In functional experiments in a tumour cell-endothelial cell interaction adherence model, adhesion was consistently affected (Fig. 7l, m), indicating that CXCL8 also upregulated ICAM1, promoting adhesion between tumour cells and endothelial cells. In vivo, anti-CXCL15 also decreased the incidence of liver metastasis in the orthotopic model (Supplementary Fig. 10g). In T1M1 PDAC cells, the levels of ICAM1 and CXCL8 were also elevated

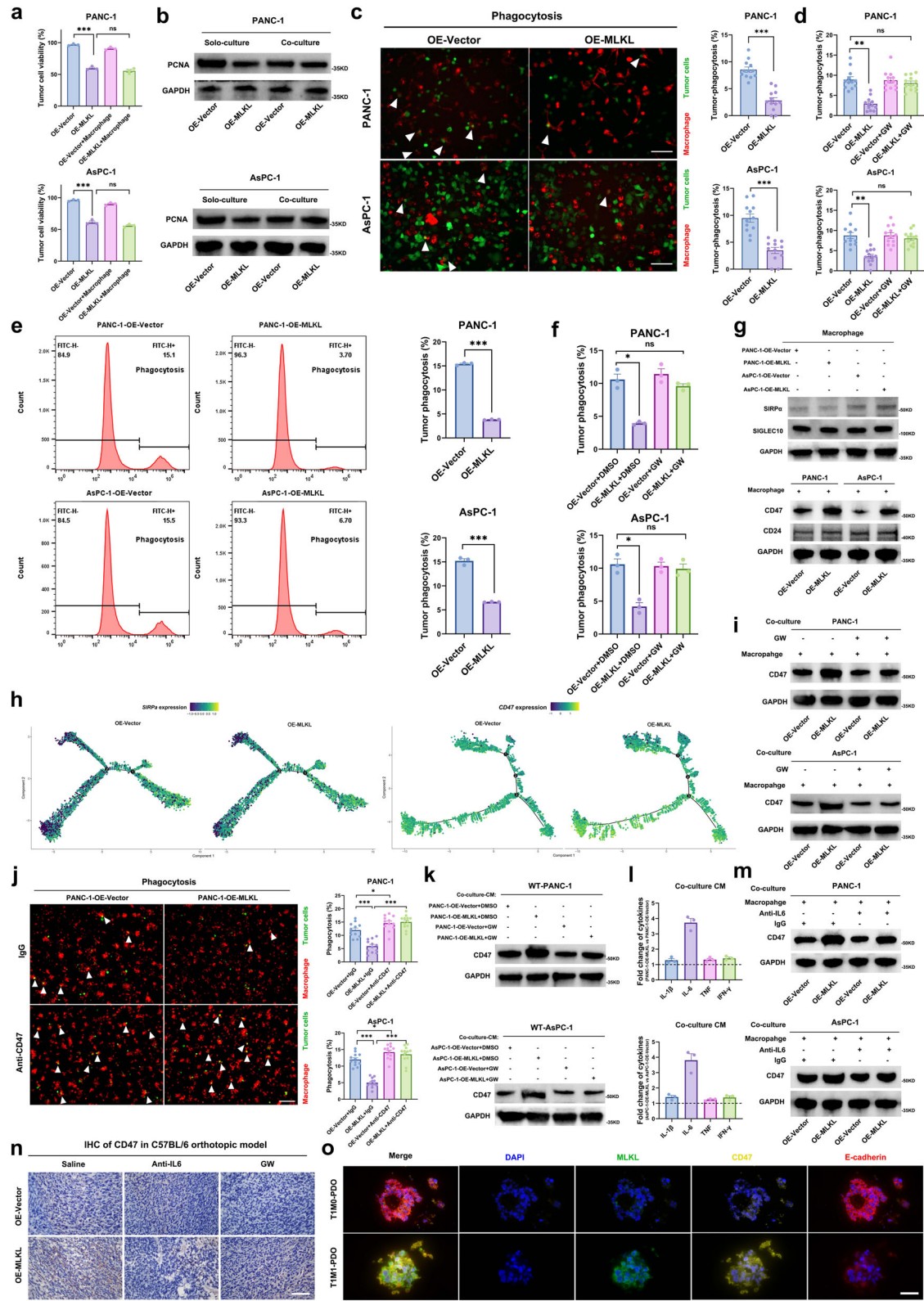

(Supplementary Fig. 10h). These results underscore the critical role of CXCL8 in multiple steps of tumour metastasis induced by MLKL-driven necroptosis.

## METs act as a scaffold that traps CXCL8 and cleave CXCL8 to form the CXCL8 monomer

Our results suggested that MET-mediated triggering of the cytokine CXCL8 plays a role in the necroptosis-mediated metastasis of PDAC by enhancing EMT and endothelial adhesion. We were interested in whether METs are associated with different forms of CXCL8, activated or nonactivated. Our ELISA and cytokine array data indicated that MLKL-driven necroptosis resulted in the release of CXCL8, which induced MET formation, and that these METs were further responsible for the increase in CXCL8 levels in the co-culture system. Considering ELIAS's detection defects for different forms of CXCL8, we developed an ELISA combined with molecular exclusion chromatography to

**Fig. 5 | MLKL-driven necroptosis induces phagocytosis resistance by enhancing tumour CD47 'don't eat me' signal. a** Cell viability assay of tumour cells cultured alone or co-cultured with macrophages. **b** WB of PCNA in tumour cells cultured alone or co-cultured with macrophages for 72 h. **c** Representative immuno-fluorescence images and quantification of tumour cell (green) phagocytosis by DiI-labelled macrophages (red) under direct co-culture. Arrows indicate phagocytic events; Scale bar: 50 μm. **d** Phagocytosis events based on immunofluorescence staining of tumour cells treated with DMSO or GW (1 μM). **e** Phagocytosis events based on flow cytometry. **f** Phagocytosis events in tumour cells treated with DMSO or GW (1 μM) determined via flow cytometry. **g** WB of SIRPα and SIGLEC10 on indirectly co-cultured (72 h) macrophages in the upper layer and CD47/CD24 on tumour cells in the lower layer. **h** Expression of CD47 in cancer cells and SIRPα in macrophages projected onto the single-cell trajectories. **i** WB of CD47 in tumour cells co-cultured with macrophages and treated/untreated with GW. **j** Phagocytosis

events were detected after treatment with IgG or anti-CD47; Arrows indicate pha-gocytosis events; Scale bar: 50 μm. **k** WB of CD47 in wild-type tumour cells treated with different co-culture CMs. **l** The foldchange of IL-1β, IL-6, TNF and IFN-γ in CMs derived from co-culture of OE-MLKL cells and control cells with macrophages through ELISA analysis. **m** WB of CD47 in tumour cells neutralised with IgG (200 μg) and anti-IL6 (200 μg) under co-culture. **n** CD47 IHC staining of primary tumours from C57BL/6 orthotopic model received different treatments: saline, anti-mouse IL6 (200 μg, iv., 3×/week), or GW (100 μM in 50 μl, iv., 3×/week); mice were eutha-nized 4 weeks after injection; *n* = 5 mice for each group; Scale bar: 50 μm. **o** Representative mIFS of T1M1-PDAC PDOs (*n* = 6) and T1M0-PDAC PDOs (*n* = 6); Scale bar: 50 μm. *n* = 3 biologically independent samples in (**a–g**), (**i–m**). Unless specified otherwise, the data are presented as means ± SEM (error bar) and com-pared using the two-sided Student's *t* test; *P < 0.05; **P < 0.01; and ***P < 0.001; ns, no significance. Source data are provided as a Source Data file.

analyse the active monomeric form of CXCL8 (Fig. 8a). Compared to that in CM derived from tumour cells cultured alone, the amount of the active form of CXCL8 (monomer) involved in MLKL-driven necroptosis was much greater in CM derived from the co-culture experi-ments (Fig. 8b).

To further confirm METs' role in increasing the active CXCL8 monomer, we introduced METi, which reduced the CXCL8 monomer level in CM (Fig. 8c). The activation of CXCL8 involves matrix metal-loproteinases (MMPs). MMPs can cleave CXCL8 into fragments of different lengths, with the 72AA form of CXCL8 considered the most biologically active. METs are associated with MMPs on the DNA scaf-fold. Therefore, we hypothesized that METs trap CXCL8 on the DNA scaffold, where it is cleaved by MMPs. To test this possibility, we used MMP inhibitors (MMP2 inhibitor, MMP2-IN-1, 10Um; MMP9 inhibitor, MMP-9-IN-1, 10 μM; MMP12 inhibitor, MMP408, 2 nM) and DNA scaf-fold digestor (Dnase I) in the co-culture system. DNase I and the MMP12 inhibitor effectively prevented CXCL8 activation (on the basis of the CXCL8 monomer level), while the MMP9 and MMP2 inhibitors had only modest effects (Fig. 8c). We performed confocal microscopic imaging and observed that a significant amount of CXCL8 adhered to the DNA network backbone released by METs and co-localized with MMP12, but just a little were co-localized with MMP2 or MMP9 (Fig. 8d, e). This further strengthens the idea that the DNA scaffold of METs provides a site for cleavage by MMP12.

The dimerization region of CXCL8 (residues 23 to 29) coincides with MMP cleavage sites (Fig. 8f). Our experiments with different MMP inhibitors validated MMP12 as a key enzyme in the cleavage of CXCL8, which prevents dimer formation and favours the monomeric form (Fig. 8g, Supplementary Fig. 11a). This finding supports our hypothesis that METs serve as a scaffold, trapping CXCL8, which is then efficiently cleaved by MET-associated MMP12, releasing CXCL8 as an active monomer that promotes metastasis.

### METs create a pro-metastatic ECM-degrading niche

METs-triggered CXCL8 plays a role in promoting crosstalk with tumours, enhancing EMT alongside other soluble cytokines and facil-itating tumour cell adherence to the endothelium. To complete the metastatic process, tumour cells must navigate through the ECM, which involves collagen degradation and MMPs activity, as indicated by KEGG and GO analyses (Fig. 7c, d). T1M1 PDAC samples showed a tumour metastasis-friendly ECM with few collagenous obstructions (Supplementary Fig. 11b). We found through gelatine zymography assays that CM derived from the co-culture experiments (MLKL-over-expressing tumour cells and macrophages) had an improved ability to digest gelatine, but this property was reversed by MMP inhibitors, MET inhibitors and GW, primarily affecting by MMP2 and MMP9 (Supple-mentary Fig. 11c). Our in vivo analysis of orthotopic tumours with picrosirius red staining also revealed a decrease in the ECM (Supple-mentary Fig. 11d), and inhibiting MMP2 or MMP9 alone was not as effective as the MET inhibitor or GW. These findings suggest that METs

establish a pro-metastatic ECM degradation niche, which promote liver metastasis through multiple steps.

### Combination treatment with GW and anti-CD47 inhibits T1M1 PDAC progression and metastasis

To further validate our findings, we performed multiplex immuno-fluorescence staining (mIFS) of 96 external PDAC patient tissues with different levels of MLKL expression in a microarray. Higher MLKL levels corresponded to lower E-cadherin levels and higher CD47 and CitH3 levels, indicating an increased 'don't eat me' signal and increased METs (Fig. 9a–c).

In in vivo experiments in which MLKL, METs, and CD47 were targeted, we observed substantial therapeutic potential (Supplemen-tary Fig. 12a–c). GW effectively inhibited the metastatic capability of KPC mouse-derived organoids co-cultured with macrophages and their ability to form METs, similar to what was observed with METi (Cl-amidine) treatment (Fig. 9d). Anti-mCD47 inhibited organoid growth effectively (Fig. 9d). Administering GW in combination with anti-mCD47 antibody resulted in the greatest significant inhibitory effect on both organoid growth and metastatic capacity, and similar results were obtained when T1M1 PDAC PDOs were co-cultured with periph-eral blood mononuclear cell (PBMC)-derived macrophages (Fig. 9e, f).

Considering that GW displayed notably encouraging performance in suppressing tumour metastatic potential when MLKL was over-expressed (Fig. 9d, e, Fig. 3g, j), we further scrutinized its off-target effects since it has also been reported to inhibit VEGFR2, a pivotal angiogenic factor[47,52,53]. We found that the overexpression of MLKL did not have a significant effect on the levels of VEGFR2 expression in orthotopic tumours (Supplementary Fig. 12d). In the orthotopic tumours, GW did not show any significant impact on angiogenesis in the OE-Vector group (Supplementary Fig. 12e). However, it was effec-tive in inhibiting the elevated angiogenesis observed in the OE-MLKL group (Supplementary Fig. 12e). Considered with the similar impacts of GW on liver metastasis tumour burden in mice from the OE-Vector group (Fig. 3g), these results suggest that GW's VEGFR2-inhibitory role is either negligible or dispensable in mitigating metastatic capacity within this context.

In terms of tumour size control, when MLKL-overexpressing KPC cells were subcutaneously transplanted into C57BL/6 mice, the com-bination regimen achieved the greatest improvement in tumour size control and survival, surpassing that of GW or CD47 blockade alone (Fig. 9g, h, Supplementary Fig. 12f). In the liver metastasis mouse model, GW showed a well performance in inhibiting liver metastasis, and the combination regimen induced the best therapeutic response, reducing liver metastasis and the primary tumour burden and enhan-cing survival (Fig. 9i, j). When the total tumour burden was assessed in C57BL/6 orthotopic model mice, the combination treatment yielded the greatest reductions in tumour burden and the incidence of liver metastasis (Fig. 9k, l). Throughout the treatments, no significant fluc-tuations in weight were observed (Fig. 9m). The combination regimen

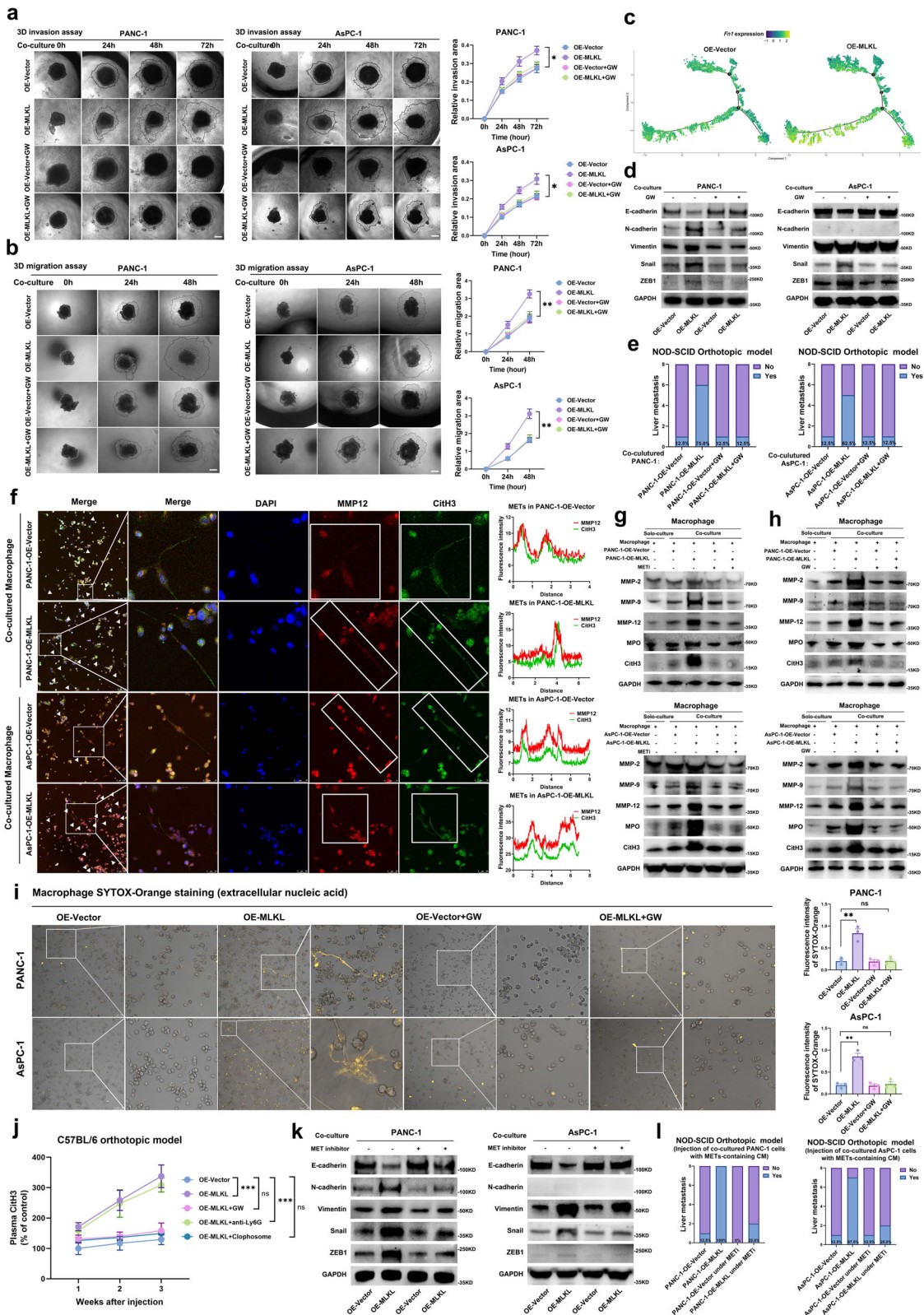

was well tolerated without noticeable systemic toxicity (Supplementary Fig. 12g).

## Discussion

Distant metastasis significantly worsens the prognosis for patients with PDAC, and liver metastases in T1M1 PDAC patients are frustrating because while surgeons could have technically removed the primary tumour, radical surgery is ruled out due to the presence of liver metastases[2]. In our study, we focused on these patients with T1M1 PDAC and have uncovered a novel immune evasion manner that we call 'don't eat me, let's run'. This strategy, which we also call "partially harmful action to promote the greater good", creates a metastatic tumour microenvironment (TME) through MLKL-driven necroptosis, stimulating the 'dont eat me' signal (CD47) and METs formation to

**Fig. 6 | MLKL-driven necroptosis induces METs formation to promote tumour metastasis. a** 3D tumour spheroid-based invasion of co-cultured tumour cells; Scale bar: 100 μm. **b** 3D tumour spheroid-based migration of co-cultured tumour cells; Scale bar: 100 μm. **c** Expression of *Fn1* in cancer cells projected onto the single-cell trajectories. **d** WB of EMT markers in indirectly co-cultured tumour cells treated with or without GW. **e** Incidence of liver metastasis in NOD-SCID orthotopic model. One million co-cultured tumour cells treated with saline or GW were injected, and mice were further treated with or without GW (100 μM in 50 μl, iv., 3×/week); mice were euthanized 4 weeks after injection; $n = 8$ mice in each group. **f** Representative immunofluorescence staining of CitH3 and MMP12 in macrophages indirectly co-cultured with tumour cells. **g** WB of MET markers in solo-cultured macrophages and co-cultured with macrophages treated with DMSO or MET inhibitor (Cl-amidine, 25 μg/ml). Macrophages were collected together with the CM as the sample. **h** The effect of GW treatment on MET-related protein levels was determined by WB analysis. **i** After indirectly coculturing with tumour cells for 72 h under different treatments, the macrophages were stained for extracellular nucleic acids with SYTOX-Orange (0.6 μM); Scale bar: 100 μm. **j** ELISA analysis of plasma CitH3 in C57BL/6 orthotopic model mice received different treatments: GW (100 μM in 50 μl, iv., 3×/week), anti-Ly6G (100 μg, iv., 3×/week), or macrophage depletion (clodronate liposomes, 200 μl, iv.); $n = 3$ mice for each group. **k** WB of EMT markers in co-cultured tumour cells treated with DMSO or MET inhibitor (Cl-amidine, 25 μg/ml). **l** Incidence of liver metastasis in NOD-SCID orthotopic model. One million tumour cells co-cultured under saline or MET inhibitor (Cl-amidine, 50 mg/kg) and the corresponding concentrated supernatant were injected. $n = 8$ mice in each group. mice were euthanized 4 weeks after injection. $n = 3$ biologically independent samples in (**a–c**, **f–h**, **j**, **l**). Unless specified otherwise, the data are presented as means ± SEM (error bar) and compared using the two-sided Student's $t$ test; *$P < 0.05$; **$P < 0.01$; and ***$P < 0.001$; ns, no significance. Source data are provided as a Source Data file.

facilitate tumour metastasis. This process initiates EMT, ECM degradation, and endothelial adhesion, ultimately promoting liver metastasis in patients with PDAC (Supplementary Fig. 13). The promising combination of MLKL and CD47 blockade can target this niche and may resolve the dilemma of liver metastasis in T1-stage PDAC patients.

As previous study did, we introduced exogenous MLKL to induce necroptosis, which partially inhibited the proliferation of PDAC cells[11,53–55]. This form of necroptosis, which we refer to as "MLKL-driven necroptosis", appears to be RIPK3 independent, in line with previous evidence[53,55,56]. While the aberrant elevation of MLKL levels does not seem to directly orient from KRAS/TP53 mutation, this unique expression pattern leads to a special form of necroptosis in T1M1 tumour cells and is quite different from the expression pattern shown by necroptosis-related molecules in the pancreatic ductal epithelium. And given that MLKL-overexpression and MLKL-knockout did not exhibit opposite effects on primary tumour size in vivo, it reflects the necroptosis-independent role of MLKL; We did not completely abolish the necroptosis-independent role and necroptosis-dependent role of MLKL through MLKL knockout, but we introduced exogenous MLKL into RIPK3[null] PANC1 and RIPK3[high] AsPC-1 cells to simulate MLKL-driven necroptosis, which both leads to the consist subsequent TME events that promote tumour metastasis[7,57–59]. This highlights the role of MLKL-driven necroptosis in this context, regardless of the RIPK3 status, which is similar to the findings of previous reports[6,53,55]. While classic RIPK3-dependent necroptosis is known to foster an immunosuppressive TME that includes M2 TAMs[22,60], we observed an unexpected increase in M1 TAMs during MLKL-driven necroptosis. However, their tumour-killing abilities were compromised by the development of phagocytosis resistance in neighbouring tumour cells through the upregulation of CD47 in the highly inflammatory microenvironment (with elevated IL-6 levels). The discovery that MLKL-driven necroptosis induces the upregulation of CD47 is novel, and we believe that this mechanism enables nearby PDAC cells to survive and metastasize away from areas of necroptosis-induced cell death[61]. The cytokine release profile during MLKL-driven necroptosis is distinct and characterized by inflammatory cytokines, unlike release of the immunosuppressive cytokines CXCL-1 and SAP-130 during RIPK3-dependent necroptosis[60].

After metastasizing to the liver, PDAC cells maintain high CD47 expression, enabling them to survive in the phagocyte-rich liver and subsequently undergo systemic metastasis[62]. The unique anti-phagocytic traits of metastatic cells in the phagocyte-rich liver provide an opportunity to target the 'don't eat me' signal (CD47), eliminating liver metastases and controlling the primary tumour by reinitiating M1 macrophage phagocytosis-mediated antigen presentation and adaptive immunity.

In addition to their resistance to phagocytosis by macrophages, surviving PDAC cells metastasize to the liver in a macrophage-dependent manner, as macrophage depletion reduces the extent of metastasis caused by MLKL-driven necroptosis. However, it is more likely that not only macrophages but also other cell types are involved in this context, due to the complex interactions between macrophages and other cell types in the process of metastasis[63–67]. This finding may also indirectly indicate that there are other cells substituted after macrophage depletion, as liver metastasis was not completely prevented after macrophage depletion. The existing evidence is not sufficient to fully identify these cell types, and further exploration is needed in the future.

Most interestingly, we observed METs formation in MLKL-driven necroptosis after ruling out macrophage necroptosis because both processes similarly release DNA and cytokines/chemokines; however, these two types of cell death are current believed to be distinct—METosis and necroptosis[26–33]. Yuan et al. reported the release of CXCL8, CXCL1, CXCL2, and CSF-2 during necroptosis, and we reported that MLKL-driven necroptosis prominently releases CXCL8, which triggers METs formation[68]. We also noted a significant increase in the levels of macrophage chemokines, such as CCL2 and CCL5, but only a modest increase in the neutrophil chemokines CXCL1/2/3[69]. Additionally, METs were found to contain MMPs that can cleave and inactivate CXCL1/2/3, potentially explaining the lack of significant difference of neutrophil infiltration. This suggests a specific TME response that METs formation during MLKL-driven necroptosis. METs formed in a background of increased M1 macrophages, which is consistent with the findings of previous studies, but we are inclined to believe that these cells may come from a unique subset/cluster of macrophages[70].

While previous studies have highlighted the role of METs in immune escape by fungi[17], our research elucidates the importance of METs in three key steps of tumour metastasis: initiating EMT, degrading the ECM, and adhering to endothelial cells during extravasation, acting as a "let's run" signal. Our findings expand the understanding of METs in tumours and the role of TAMs in promoting metastasis. METs may act similar to NETs and form an extracellular barrier that hinders immune cell infiltration or induce pro-tumour signals in tumour cells through MMPs[16]. In our study, MLKL-driven necroptosis increased the levels of various EMT-enhancing cytokines in the TME, and METs were the main factor responsible for the increase in CXCL8 levels and partially contributed to the increases in IL-6 and GM-CSF levels.

Interestingly, we found that the DNA scaffold of METs trapped CXCL8 and processed CXCL8 through the cleavage activity of MMP12, which is a characteristic component of METs[71]. Processing by MMP12 prevents CXCL8 from forming a dimer and leads to an activated CXCL8 monomer[72]. In addition to the well-known angiogenic effects of activated CXCL8, it also enhances the expression of ICAM1 in the context, which aids tumour cell adhesion to endothelial cells and liver metastasis[47,49–51,73]. METs also effectively remodel the ECM, creating a pro-metastatic niche, partly due to the ability of its pivotal component —MMP12 to broadly activate other MMPs[74].

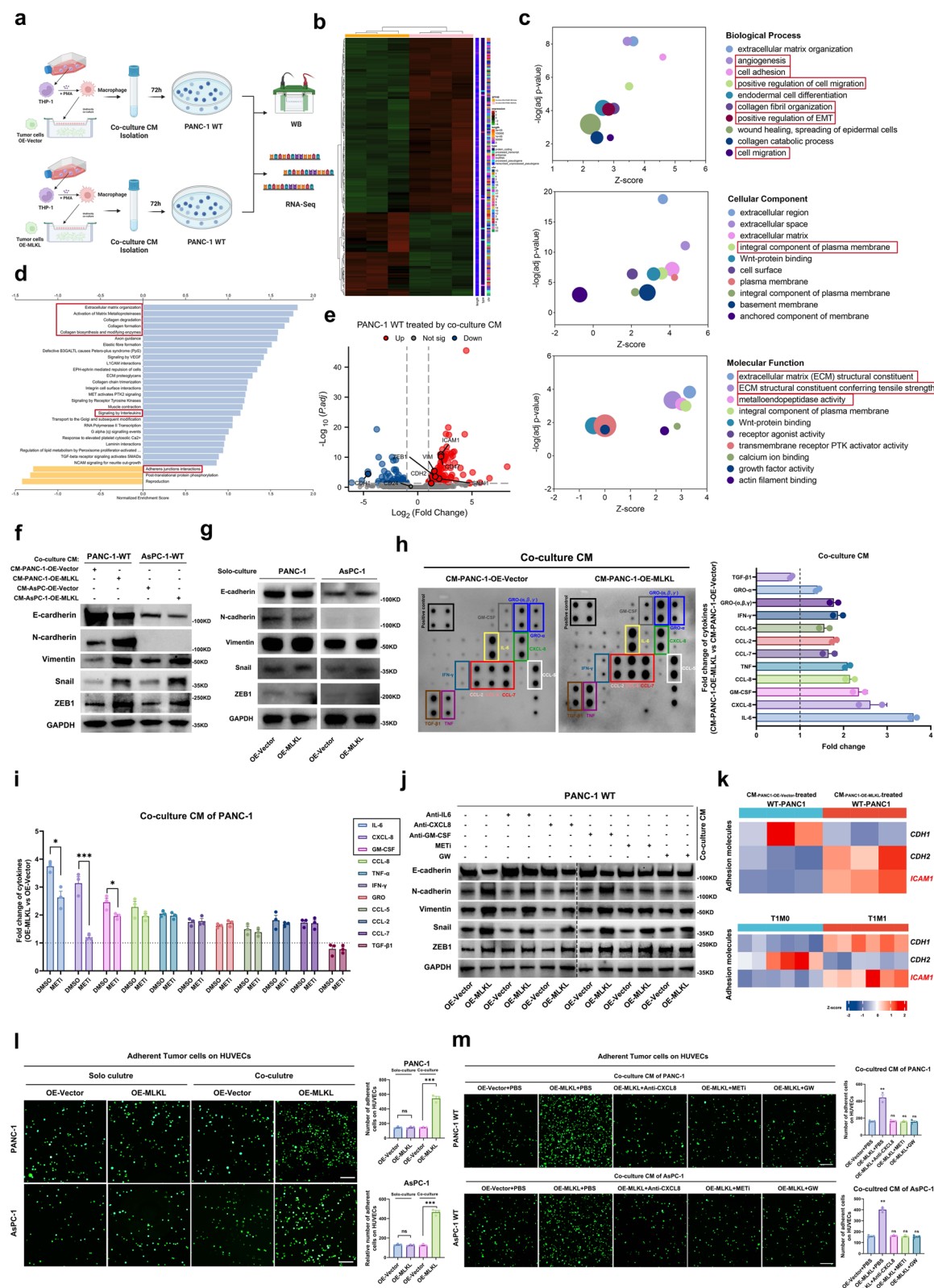

We examined T1M1 PDAC PDOs and MLKL-driven necroptosis in vivo using preclinical models and investigated the potential of GW, MET inhibitor (Cl-amidine) and anti-CD47 for restoring the opportunity for radical resection. GW in combination with anti-CD47 therapy effectively blocked primary foci metastasis, eliminated existing liver metastases, and controlled primary tumours. The necroptosis inhibitor - GW is also reported to inhibit VEGFR2[52,53], our results did not

substantiate that it was its VEGFR2-inhibitory role that decreased the metastatic capacity enhanced by MLKL-driven necroptosis. Interestingly, GW showed a favourable effect in inhibiting angiogenesis of OE-MLKL tumours, which might be attributed to its necroptosis-inhibitory role and thereby mitigated the activation of CXCL8, a cytokine known to promote angiogenesis[47]. This further support the penitential of GW for PDAC with overexpressed MLKL. Cl-amidine also showed

**Fig. 7 | METs promote metastatic capacity through the CXCL8/CXCR pathway, promoting the EMT and endothelial adhesion capacity of tumour cells.** **a** Workflow showing the treatment of wild-type PANC-1 cells for subsequent WB analysis and RNA-seq. **b** Heatmap showing the DEGs in PANC-1 cells treated with different co-culture CMs ($n = 3$ in each group). **c** GO analysis of DEGs. **d** KEGG pathway analysis of DEGs. **e** Volcano plot of DEGs. **f** WB of EMT markers in wild-type PANC-1 and AsPC-1 cells treated with different co-culture CMs; $n = 3$ biologically independent samples. **g** WB of EMT markers in tumour cells culture alone; $n = 3$ biologically independent samples. **h** Representative human cytokine antibody array and the corresponding quantitation of co-culture CM; $n = 3$ biologically independent samples. **i** ELISA of cytokines of CM in co-culture system treated with DMSO or MET inhibitor (Cl-amidine 25 μg/ml); $n = 3$ biologically independent samples. **j** WB of EMT markers in wild-type tumour cells treated with different CMs collected following co-culture system (anti-IL6, 200 μg; anti-GM-CSF, 200 μg; anti-CXCL8, 200 μg; MET inhibitor, Cl-amidine, 25 μg/ml; GW, 1 μM) for 48 h; $n = 3$ biologically independent samples. **k** Expression matrix of adhesion molecules in wild-

type tumour cells that received different CMs ($n = 3$ each group), and in T1M0-PDAC ($n = 6$) and T1M1-PDAC ($n = 6$) according to RNA-seq data. **l** GFP-labelled tumour cells co-cultured with macrophages for 48 h were subjected to a HUVEC adhesion assay. Adherent tumour cells are indicated in green; $n = 3$ biologically independent samples; Scale bar:100 μm. **m** GFP-labelled wild-type tumour cells received different CMs collected from co-culture system under different treatments (anti-CXCL8, 200 μg; MET inhibitor, Cl-amidine, 25 μg/ml; GW, 1 μM) for 48 h h and were subjected to a HUVEC adhesion assay. Adherent tumour cells are indicated in green; Adherent tumour cells in each group were compared with those in the OE-Vector+PBS group; $n = 3$ biologically independent samples; Scale bar:100 μm. Unless specified otherwise, the data are presented as means ± SEM (error bar) and compared using the two-sided Student's $t$ test; *$P < 0.05$; **$P < 0.01$; ***$P < 0.001$; ns, no significance. **a** Created with BioRender.com released under a Creative Commons Attribution-NonCommercial-NoDerivs 4.0 International license (https://creativecommons.org/licenses/by-nc-nd/4.0/deed.en). Source data are provided as a Source Data file.

somewhat efficacy, underscoring the importance of METs and the complexity of MLKL-driven necroptosis-mediated immune evasion. The development of more targeted MET inhibitors is promising[75]. At this stage, GW and anti-CD47 therapy holds potential clinical value, although further research into the mechanisms of MLKL activation is needed. Furthermore, further specific research on the reasons for the aberrant increase in MLKL levels in T1M1 PDAC and the interactions between MLKL-driven necroptosis/METs and other immune cells are needed.

In summary, our research has revealed the role of MLKL-driven necroptosis in early T-stage PDAC with liver metastasis, involving phagocytosis resistance, METs formation and creating a pro-metastatic niche. We suggest a promising combination therapeutic regimen comprising GW and anti-CD47 to address the clinical challenge of treating early T-stage PDAC with liver metastasis, aiming to enable radical surgery.

# Methods
## Cell lines
Human PDAC cells (PANC-1, CRL-1469; AsPC-1, CRL-1682) and myeloid cell lines THP1 (TIB-202) and RAW 264.7 (TIB-71) were obtained from American Type Culture Collection (ATCC, Manassas, VA, USA) and were cultured in standard media per ATCC recommendations that was supplemented with heat-inactivated FBS and antibiotics (50 U/mL penicillin and 50 mg/L streptomycin) in a humidified atmosphere at 37 °C containing 5% $CO_2$. HUVEC cells (CL-0675) were purchased from Pricella Life Science & Technology Co., Ltd and cultured with Complete culture medium for PUMC-HUVEC-T1 (Pricella Life Science & Technology Co., Ltd, CM-0675) at 37 °C containing 5% $CO_2$. All cell lines were tested to confirm that they were free of mycoplasma and authenticated by short-tandem repeat analysis.

## Cell transfection
MLKL lentiviruses or plasmids were purchased from GeneChem (Shanghai, China). Lentivirus transfection was performed by Gene-Chem (Shanghai, China) and plasmids were transfected into cells using the Roche Transfection Reagent (Roche, 578 Basel, Switzerland).

## Macrophage generation and stimulation
Human THP-1 cells that were in the logarithmic growth phase were seeded at $1.0 \times 10^5$ cells/mL in 80 ng/mL phorbol-12-myristate 13-acetate (PMA, MedChemExpress, HY-18739) for 48 h and allowed to differentiate into adherent THP-1-derived un-activated macrophages (M0) for subsequent experiments. For co-culture assays with PDAC cells, THP-1 derived macrophages were seeded in the upper or lower layer of a trans-well plate according to the purposes.

## Preparation of human macrophages from peripheral blood monocytes
Human peripheral blood monocytes (PBMCs) were isolated from buffy coats by Hypaque-Ficoll density gradient centrifugation (Solarbio, P8900). PBMCs were differentiated into macrophages by cultivation in RPMI-1640 with 25 ng/mL M-CSF (R&D, 216-MC).

## Culture and expansion of pancreatic cancer organoids
Pancreatic cancer organoids were derived from tumour tissues of patients with stage T1M1 or T1M0 pancreatic cancer. The tumour tissues were washed in phosphate-buffered saline (PBS) (Gibco, 10010023), cut into small pieces (2-3 mm) using sterile scalpels, and digested in a MasterAim tissue enzymatic solution I (AIMINGMED, 100-050) in Advanced DMEM/F12 medium (Gibco, 12634010) at 37 °C for 1 h with intermittent shaking. Additional digestion was performed using the MasterAim tissue enzymatic solution I (AIMINGMED, 100-051). Tissue digestion was stopped by adding Advanced DMEM/F12 that contained 10% FBS. A total of 500 cells were resuspended in 20 μL of Matrigel (Corning, 356231), and cells were then plated in individual wells of a 48-well plate and incubated at 37 °C for 5 min to allow solidification. Cells were cultured in a basic medium (advanced DMEM/F12, 10 mM HEPES, 1× GlutaMAX-I, 100 μg/mL Primocin, 1× penicillin/streptomycin solution) or a complete medium (advanced DMEM/F12, 10 mM HEPES, 1× GlutaMAX-I, 100 μg/mL Primocin, 1× penicillin/streptomycin solution, 500 nM A83-01, 10 μM Y-27632, 1.56 mM N-acetylcysteine, 10 mM nicotinamide, 10 ng/mL FGF10, 1× B27 supplement, 10 μM forskolin, 30% Wnt3A conditioned medium, 2% R-spondin conditioned medium, 4% Noggin conditioned medium). Culture media were changed every 3 days.

## PDO Fixation and staining
For immunostaining, the day-12 organoids were harvested from Matrigel by scraping with a pipette tip, washed in DPBS (Gibco, 14190250), and fixed in 4% paraformaldehyde (PFA) (Electron Microscopy Sciences, 157-4-100) for 20 min at room temperature. Following fixation, organoids were gently pelleted and overlaid with 200 μL of low melting agarose (Lonza, 50101) at 42 °C and permitted to solidify. The agarose plugs containing organoids were embedded in paraffin and 5-μm sections were prepared. The sections were deparaffinized in xylene followed by rehydration using a series of graded ethanol solutions. Antigen retrieval was performed using pH 6.0 sodium citrate buffer (Thermo Fisher Scientific, 005000). Slides were washed with TBS containing 0.1% Tween 20 and incubated with MLKL (1:200), E-cadherin (1:200) and CD47 (1:200) antibodies overnight at 4 °C. Sections were washed and incubated with Alexa Fluor secondary antibodies for 1 h at room temperature and cover slipped using VEC-TASHIELD antifade mounting media with DAPI (Vector Laboratories, H-2000). Confocal microscopy images were recorded.

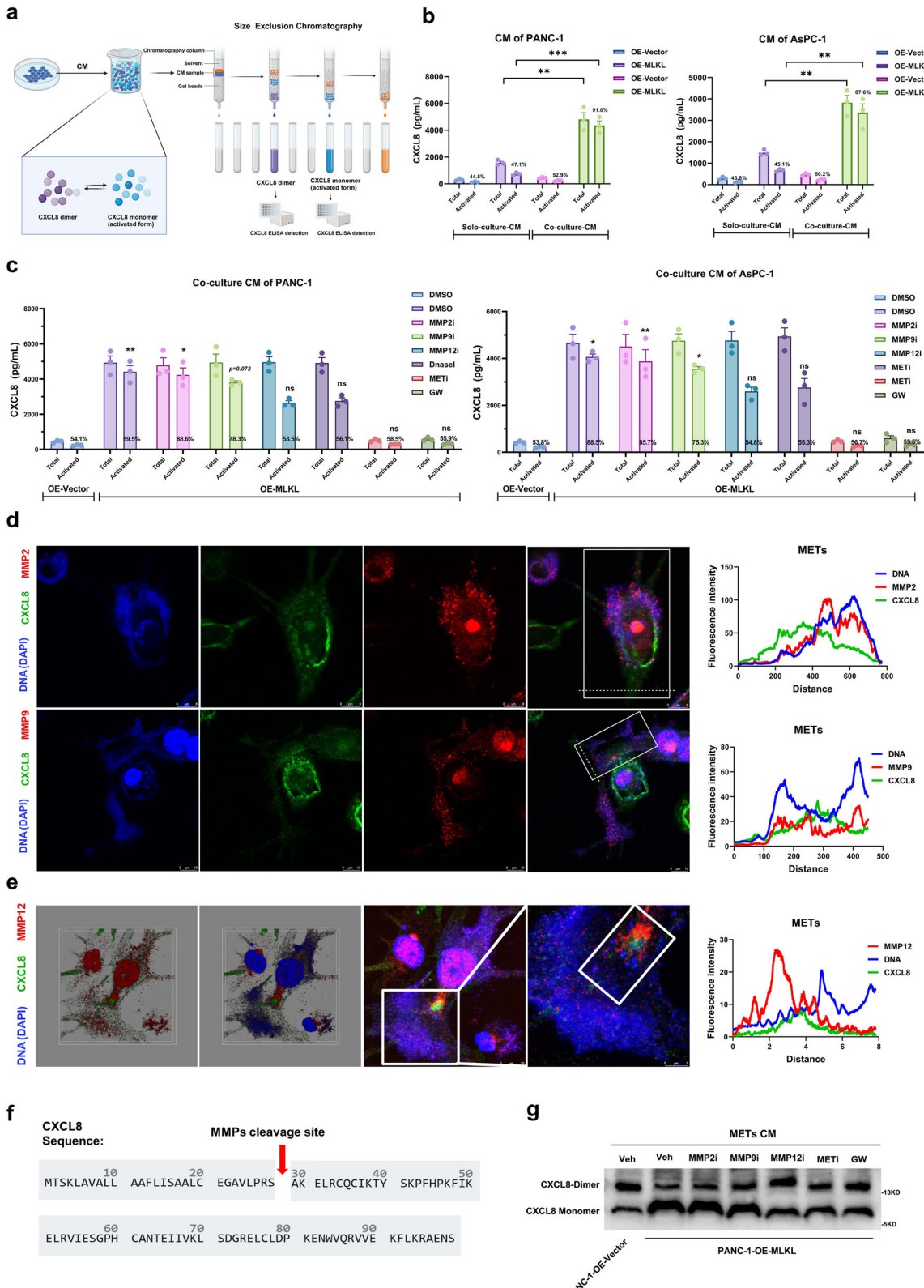

### PDO necroptosis inhibitor screening assay

For PDO phagocytosis, the day-12 organoids were harvested from Matrigel by scraping with a pipette tip as a single cell suspension, washed in DPBS (Gibco, 14190250), and then distributed evenly into each well of 96-wells plate. At 24 h after passaging, when the organoid re-formed and the bubble was round, necroptosis inhibitors: Nec-1 (1 μM, MedChemExpress, HY-15760), GSK872 (6 μM, MedChemExpress, HY-101872), or GW (1 μM, MedChemExpress, HY-112292) was added in the solitary organoid well[53,76,77]. The size was observed by microscopy for 4 consecutive days. Growth was quantified as the relative volume normalized to the volume at 24 h. The organoid volume was determined as $4/3 \times \pi \times r^3$.

**Fig. 8 | METs trap CXCL8 and cleave CXCL8 into activated CXCL8-monomer.**
**a** Workflow of size exclusion chromatography tandem ELISA-based detection of the CXCL8 monomer and dimer. **b** ELISA to quantify total and activated CXCL8 (monomer) secreted in the indicated CM ($n = 3$ each group); the CXCL8 level was compared by using two-sided Student's $t$ test; the mean percentage of activated CXCL8 (monomer) in each group was indicated; $n = 3$ biologically independent samples. **c** Quantification of total and active CXCL8 (monomer) secreted in the indicated CM after different treatments were applied ($n = 3$); the mean percentage of activated CXCL8 (monomer) was indicated and compared with the DMSO-treated ones using the two-sided Student's $t$ test; $n = 3$ biologically independent samples. **d** Cellular images and colocalization of CXCL8 (green), MMP2/MMP9 (red) and the DNA scaffold (blue) of METs; $n = 3$ biologically independent samples.

**e** Cellular images and colocalization of METs with trapped CXCL8 (green) and MMP12 (red) in the DNA scaffold (blue); $n = 3$ biologically independent samples. **f** The amino acid sequence of CXCL8 and the cleavage site utilized by MMPs. **g** WB analysis of the CXCL8 dimer and monomer from the MET-containing CM following different treatments, 20% nonreducing SDS–PAGE was used to separate the proteins; $n = 3$ biologically independent samples. Unless specified otherwise, the data are presented as means ± SEM (error bar) and compared using the two-sided Student's $t$ test; *$P < 0.05$; **$P < 0.01$; and ***$P < 0.001$; ns, no significance. **a** Created with BioRender.com released under a Creative Commons Attribution-NonCommercial-NoDerivs 4.0 International license (https://creativecommons.org/licenses/by-nc-nd/4.0/deed.en). Source data are provided as a Source Data file.

## Generation of 3D-tumor spheroids and growth assay

Tumour cell monolayers were washed twice with PBS (Gibco, 10010023), and a cell dissociation enzyme (Gibco, 25200072) was then added to obtain single cell suspensions without clusters. Then, cells were counted using a haemocytometer and diluted to a concentration of 0.5 to $2 \times 10^3$ cells/mL, with the optimal cell density determined for each cell line. The cell suspension was then transferred into a sterile reservoir and a multichannel pipette was used to dispense 200 μL/well into an ultra-low attachment (ULA) 96-well round-bottom plate (2000 cells each well). These plates were transferred to an incubator (37 °C, 5% $CO_2$, 95% humidity) to allow growth. Changes in spheroid size were determined using a high content cell quantitative imaging analysis system over 4 consecutive days.

## 3D- tumour spheroid invasion assay

After 4 days, the formation of 3D-tumor spheroids was visually confirmed before initiation of the invasion assay. Samples of 4-day old spheroids were placed in ULA 96-well plates on ice. Then, 100 μL/well of growth medium was carefully removed from each well and 100 μL of basement membrane-like matrix (Corning, 354236) was gently added and dispersed into the U-bottom well. Then, the plate was transferred into an incubator at 37 °C to allow solidification. One hour later, 100 μL/well of complete growth medium was added. Images were recorded from 0 h to 72 h[78]. The relative invasion area to the initial tumour spheroid area was quantified.

## 3D-tumor spheroid migration assay

As above, 4 days were allowed for the formation of 3D-tumor spheroids, and their formation was visually confirmed before initiation of the migration assay. Then, spheroids were gently transferred to a gelatin-precoated migration 96-well plate. Complete growth medium was added. Then, the spheroids were allowed to adhere to the coated surface for 60 min, followed by recording of images from 0 h to 48 h[79]. The relative migration area to the initial tumour spheroid area was quantified.

## In vitro phagocytosis assay

Pretreated THP-1-derived macrophage were mechanically detached and labelled with 5 μM DiI (Beyotime, C1036), then seeded into a transparent 24-well tissue culture plate at $5 \times 10^4$ cells/well. GFP-labelled PDAC cells were harvested as a single cell suspension and seeded at $5 \times 10^4$ cells/well. For *E. coli* phagocytosis, 5 μL of *E. coli* that was labelled with GFP (Phagocytosis Assay Kit, Abcam, ab235900) were seeded. For tumour phagocytosis, tumour cell suspensions were incubated were washed and co-incubated with macrophages for 2 h. Four random fields were assessed in each replicate using fluorescence microscopy, and the number of phagocytic events was scored and averaged for each replicate.

For flow cytometry-based in vitro phagocytosis assays, tumour cells expressed virally GFP and macrophages were co-cultured in ultra-low-attachment 96-well U-bottom plates (Corning, 4515) in serum-free RPMI (Gibco, 11875119). Plates were washed two times; macrophages were added to the plate; and plates were then incubated for 2 h at

37 °C. Phagocytosis was analysed by flow cytometry, and measured as the number of APC-CD45+ (Invitrogen, 17-9459-42), PE-CD11b+ (Proteintech, PE-65116) and GFP+ macrophages as a percentage of the total CD11b+ macrophages. For further treatment, IgG or Anti-CD47 at a concentration of 10 μg/mL were incubated with tumour cells for 20 min in a humidified 5% $CO_2$ incubator at 37 °C before co-incubated with macrophages[80].

## Flow cytometry identification of macrophages

Harvested single-cell suspensions of macrophages were first treated with an FcR-blocker (BD Pharmingen, 564219) for 10 min at 4 °C. Then, different antibodies purchased from Invitrogen (FITC-CD-80, 11-0809-42; APC-CD86, 17-0869-42; PeCy7-CD163, 25-1639-42; PE-CD206, 12-2069-42) were added. A BD Accuri C6 was used for flow cytometry analysis, and FlowJo software was used to analyse the data.

## Phalloidin staining

Cells were fixed in 4% formalin for 20 min, washed three times with PBS, and stained with 5 μg/mL of phalloidin conjugate solution (Invitrogen, A12381) in PBS for 40 min at 37 °C. The cells were washed three times with PBS to remove the unbound phalloidin conjugate and imaged by confocal microscopy. Cells with extensively-formed extended filopodia and lamellipodia indicate stronger invasive capacity; Cells exhibiting loss of spindle-shaped features and fewer filopodia and lamellipodia indicates less invasive capacity. At the 3D culture condition, after co-culture with macrophage, we planted co-cultured tumour cells on 3D low-attachment plates, tumour spheroid exhibiting irregular shapes with pseudopodia-like extensions indicates stronger invasive capacity. as evidenced by analyses of single-plane images, Z-stack images, 0.8um

## METs quantification using SYTOX-Orange

Macrophages (100,000) were seeded onto 6-wells plates and indirectly co-culture with tumour cells or CM (with vehicle or cytokines and/or inhibitors as indicated) for 8 h. 50 nM SYTOX-Orange (Invitrogen, S11368) was then added to the plate and after 5 min, images were recorded by confocal microscopy and fluorescence intensity was measured at excitation/emission wavelengths of 547/570 nm.

## Conditioned media preparation

Cancer cells were grown in complete medium, washed twice with PBS, and subsequently incubated at 37 °C as indicated treatments. After 48 h, culture medium was collected, centrifuged at 500 $g$ for 5 min to remove cell debris and the supernatant were stored at −80°C. METs-contained culture medium was collected from co-culture experiments (MLKL overexpressing tumour cells and macrophages) and concentrated via Amicon Ultra-4 Centrifugal Filter Devices (Millipore, UFC801008D) for further use.

## Plasma sample collection from mice

Plasma samples were collected from cardiac blood using a syringe with a 25 G needle and placed into tubes with ethylenediaminetetraacetic

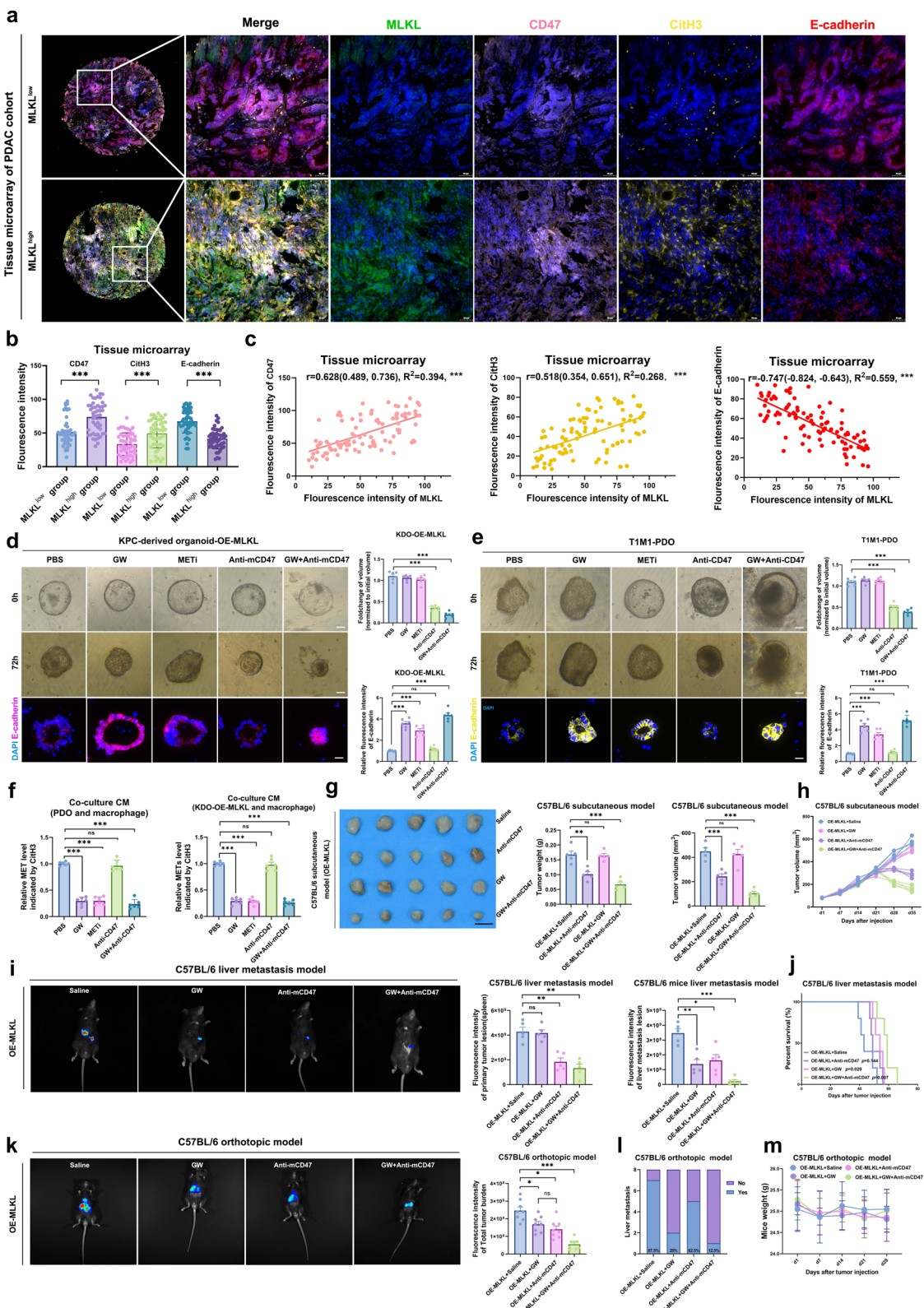

acid (EDTA). Whole blood was centrifuged at 4 °C at 1300 g for 10 min, and the top plasma layer was collected.

### Immunofluorescence of METs and cell culture

Cells were fixed with 4% PFA (Electron Microscopy Sciences, 157-4-100) for 20 min. After fixation, they were rinsed twice in PBS, incubated in 50 mM of NH₄Cl (MedChemExpress, HY-Y1269) for 10 min and

permeabilized with 0.5% Triton X-100 (Thermo Scientific, 85111) for 5 min. Cells were next blocked in PBS containing 1% bovine serum albumin (BSA) (Thermo Scientific, 37520) for 60 min and incubated with anti-CitH3 (1:200, Abcam, ab281584), anti-MMP12 (1:400, HUA-BIO, ET1602-42) antibodies in blocking buffer overnight at 4 °C. After two washes in PBS, cells and matrices were incubated in the presence of fluorochrome-conjugated secondary antibodies (1:250) for 40 min,

**Fig. 9 | GW combined with anti-CD47 suppresses the progression and metastasis of T1M1-PDAC. a**–**c** mIFS, quantification and correlation analysis of E-cadherin, MLKL, CD47 and CitH3 in external PDAC array (*n* = 96); Correlation analysis was done through the Pearson correlation coefficient (*r*). *r* was represented with 95% confidence intervals; effect sizes: $R^2$; degrees of freedom: 94, two sided; ***$P < 0.001$; Scale bar: 50 μm. **d**, **e** Representative images of KDOs and T1M1-PDOs received different treatments and quantitation of organoid volume and E-cadherin fluorescence; *n* = 6 biologically independent samples; Scale bar: 50 μm. **f** Quantitation of METs (CitH3) by ELISA in KDO CM (*n* = 6) and T1M1-PDAC PDO CM (*n* = 6). **g** Tumours gross morphology, weight and volume of C57BL/6 mice received different treatments: saline, GW (100 μM in 50 μl, iv., 3×/week), anti-mCD47 (400 μg, ip., 3×/week) or both since day 21 after injection ($10^6$ KPC OE-MLKL cells); Tumours were harvested at five weeks after injection (5 mice per group). **h** Tumour volume curve of C57BL/6 mice that received different treatments since day 21 after injection ($10^6$ KPC OE-MLKL cells); (5 mice per group). **i** Bioluminescence of liver metastases and spleen nodes of C57BL/6 liver metastasis model received different treatments: saline, GW (100 μM in 50 μl, iv., 3×/week), anti-mCD47 (400 μg, ip., 3×/week) or both since day 7 after injection ($10^6$ KPC OE-MLKL cells). Images were collected three weeks after injection; 5 mice per group. **j** Kaplan–Meier survival curves of C57BL/6 liver metastasis model received different treatments: saline, GW (100 μM in 50 μl, iv., 3×/week), anti-mCD47 (400 μg, ip., 3×/week) or both since day 7 after injection ($10^6$ KPC OE-MLKL cells), 5 mice per group; log-rank test. **k** Bioluminescence of total tumour burden in C57BL/6 orthotopic model received different treatments: saline, GW (100 μM in 50 μl, iv., 3×/week), anti-mCD47 (400 μg, ip., 3×/week) or both since day 7 after injection ($10^6$ KPC OE-MLKL cells); Images were collected at four weeks after injection; (8 mice per group). **l**, **m** Incidence of liver metastasis and weight in orthotopic model (8 mice per group) received different treatments; Mice were euthanized four weeks after injection. Unless specified otherwise, the data are presented as means ± SEM and compared using the two-sided Student's *t* test; *$P < 0.05$; **$P < 0.01$; and ***$P < 0.001$; ns, no significance. Source data are provided as a Source Data file.

rinsed twice in PBS, stained with 4′,6-diamidino-2-phenylindole (DAPI) for 5 min, rinsed in water, and the coverslips were mounted onto glass slides using mounting media. Confocal microscope was used for the acquisition of images and cell 3D reconstruction.

## Tumour cell-endothelial cell interaction adherent assay

Tumour cell-endothelial cell interaction adherent assay was performed as described previously. Briefly, HUVECs were seeded on 24-well plates and grown overnight to reach full confluence. Then, $5 \times 10^5$ GFP-labelled tumour cells of control or MLKL-overexpression pretreated with macrophages were washed with PBS for three times, and re-suspended in serum-free culture medium to add onto the monolayer HUVECs and co-cultured at 37 °C for 30 min. After washing by PBS softly for three times to remove the non-adherent cells, the adherent tumour cells on the monolayers were observed and imaged with a fluorescence microscope.

## Gelatin zymography

Proteins of co-culture conditioned medium after concentrating through Amicon Ultra-4 Centrifugal Filter Devices in 2× sample buffer (25% 0.5 M Tris/HCl [pH 6.8], 20% Glycerol, 10% SDS, 0.1% bromphenol blue, 20 μg) were loaded in 10% SDS-PAGE gels containing 0.1% gelatin (porcine skin, Fluka). Proteases were renatured (2.5% Triton X-100) and developed for 24 h at 37 °C (50 mmol/L Tris, 5 mmol/L $CaCl_2$, 0.2 mol/L NaCl, and 0.02% Brij). Gel-Staining with Coomassie blue staining solution (0.5% Coomassie R250, 50% MeOH, 20% acetic acid) was followed by destaining (40% MeOH, 10% acetic acid)[81].

## Size exclusion chromatography tandem ELISA-based CXCL8 monomer/dimer detection

Culture medium was collected as prior reported and were diluted in elution (PBS), 50 mM sodium phosphate containing 150 mM sodium chloride, pH 7.4) for detectable sample preparation. Prepared samples (10 μL) were injected on the 1260 Infinity II Preparative LC System (Agilent) according to the manufacturer's protocol. Conditions Column: Agilent AdvanceBio SEC 300 Å, 7.8 × 300 mm, 2.7 μm (Agilent, PL1180-5301); Mobile phase: Phosphate buffered saline (PBS), 50 mM sodium phosphate containing 150 mM sodium chloride, pH 7.4; Thermostatted Column Compartment: Ambient; Inj vol: 10 mL; Flow rate: 0.8 mL/min; Detection: UV 220 and 280 nm; The Size Exclusion Chromatography column was calibrated by measuring the elution volumes of Protein SEC Standards (Calibration kit Protein, nominal Mp 243 − 670,000 Da, Agilent, PSS-PROKITR1: Vitamin B12 PSS-pro1.4k, Aprotinin PSS-pro6.5k, Cytochrome C PSS-pro12k, Myoglobin PSS-pro17.5k, B-Lactoglobulin PSS-pro35k, Albumin from Chicken PSS-pro44k, Albumin from Bovine PSS-pro67k, Gamma-Globulins PSS-pro158k, Thyroglobulin PSS-pro670k). The log molecular weight values of the standards were plotted against the elution volume to determine the equivalent molecular weight of the sample. Mobile phase (10 μL) was injected as blank, followed by six replicates of samples to calculate area and retention time (RT) deviation. The column was calibrated using a series of standard proteins with known molecular weights. Standard protein aggregates (void peak) in the protein marker were used to calculate the void volume, which eluted at $t_0$ minutes on the column, corresponding to $V_0$ (mL). The calibration curve for proteins separated on the column shows a linear relationship, and defines the exclusion limit (670 kDa) for the protein range (1.3 to 670 kDa) analysed. during sample loading, determine the elution time of CXCL8 monomer (between Apotinin PSS-pro6.5k and Cytochrome C PSS-pro12k) and dimer (between Cytochrome C PSS-pro12k and Myoglobin PSS-pro17.5k) based on the calibration curve, and collect the corresponding elution respectively for subsequent CXCL8 ELISA kit (Abcam, ab214030) detection. Then calculate the concentrations of CXCL8 monomer and dimer in culture medium, respectively.

## Mouse experiments

The 6–8-week-old NOD-SCID mice (NOD.Cg-Prkdc^scid/NifdcSmoc, Cat. NO. SM-019) and wild-type C57BL/6 J mice (Cat. NO. SM-001) were purchased from Shanghai Model Organisms Center, Inc. The 8-week-old KPC mice (C57BL/6 Smoc-*Trp53*^em4(R172H)*Kras*^em4(LSL-G12D)Tg(Pdx1-cre)Smoc, Cat. NO. NM-KI-210096) were purchased from Shanghai Model Organisms Center, Inc. All mice used in this study were age-matched littermates. Both male and female mice were used in this study. All mice were housed and maintained under specific pathogen-free (SPF) conditions in the Anburui BD Laboratory (Fuzhou, China). The number of mice in each experiment was indicated in the corresponding figure legend. All animals were handled strictly according to the Principles for the Utilization and Care of Vertebrate Animals and the Guide for the Care and Use of Laboratory Animals. All animal experiments were approved by the Institutional Animal Care and Use Committee of Fujian Medical University (IACUC FJMU 2022-0800). For animal studies, the mice were earmarked before grouping and randomly separated into groups by an independent person, and experimental/control animals were bred separately. The number of mice in each group has been indicated in the corresponding figure legend. At the end of animal experiments, the mice were euthanised by cervical dislocation. For orthotopic tumour model, 6 to 8 weeks old C57BL/6 mice were injected with one million KPC cells that were isolated from KPC mice that had mice MLKL overexpression or control mice in a total volume of 50 μL. The cells were injected into the parenchyma of the pancreas. Three tumours in OE-vector of group and three tumours of OE-MLKL group in orthotopic pancreatic tumour model was harvested to undergo single-cell RNA-Seq and immune cell infiltration. Liver

metastasis was evaluated using micro-CT per week since 7 days after injection. Total tumour burden of the mice was evaluated using in vivo luciferase-based non-invasive bioluminescence imaging at four weeks after injection, and then, the mice were euthanized to further observe the occurrence of liver metastasis, and tumours were collected for further analysis as needed. The treatments were as indicated in the corresponding figure legends. For subcutaneous tumour model, the corresponding cells ($10^6$) were resuspended in a 1:1 solution of PBS and Matrigel in a final volume of 100 μL, were then implanted subcutaneously in the flanks of 6- to 8-week-old C57BL/6 mice. Tumour volume was determined each week after injection using the electronic callipers to measure length (L) and width (W): 0.52 × L × W × W. The mice received different treatments since beginning on day 21 after the subcutaneous injection, and the mice were euthanized at five weeks after subcutaneous injection. The treatments were as indicated in the corresponding figure legends. For liver metastasis model, the 6 to 8-week-old NOD-SCID or C57BL/6 mice received intrasplenic injections of one million KPC cells that overexpressed mice MLKL or control luciferase-labelled KPC cells to establish a metastatic liver model. The treatments were as indicated in the corresponding figure legends. Established liver metastasis was evaluated by in vivo luciferase-based non-invasive bioluminescence imaging at 3 weeks after intrasplenic injection, and then, the mice were euthanized, and tumours were collected for further analysis as needed. The maximum size of the tumours should not exceed 2 cm according to our institutional ethical board, and we have adhered to the tumour size limits in the experiments. Early termination criteria were as follows: (1) the maximum cumulative tumour burden of 2.0 cm in diameter; (2) the tumour impedes eating, urination, defecation, or ambulation; and (3) very poor body condition.

## Human tissues

Human PDAC tissues from Fujian Provincial Hospital were used for IHC, generation of PDOs, and RNA-Seq. PDAC tissue microarray sections (Shanghai Outdo Biotech Co., Ltd.) were used for multiplex IHC. This study was authorized (K2021-03-046, K2023-03-023) by the Ethics Committee of Fujian Provincial Hospital.

## mIFS and imaging analysis

The pancreatic cancer tissue array (HPanA120PG01, Shanghai Outdo Biotech Co., Ltd.) were performed mIFS using an Opal 7-colour fluorescent IHC kit (PerkinElmer) combined with automated quantitative analysis (AQUA; Genoptix) to assess the expression of MLKL, CD47, CitH3 and E-cadherin. First, the concentration and the order of the four antibodies were optimized, and the spectral library was built based on the single-stained slides. Slides were dewaxed and rehydrated through a series of xylene-to-alcohol washes before being incubated in distilled water. mIFS was then performed after heat-induced antigen retrieval. Primary antibodies to the following antigens were used: MLKL (1:1000), CD47 (1: 1000), CitH3 (1:1000) and E-cadherin (1:1000). The following secondary antibodies were used: anti-mouse Envision HRP (Dako), anti-rabbit Envision HRP (Dako) and DAPI. Fluorescence images were acquired using the Vectra 2 Intelligent Slide Analysis System using Vectra software v2.0.8 (PerkinElmer).

First, monochrome imaging of the slide at 4× magnification using DAPI was conducted. An automated algorithm (developed using in Form software) was used to identify areas of the slide containing tissue to create RGB (red-green-blue) images. Accepted images were processed using AQUAduct (PerkinElmer), wherein each fluorophore was spectrally unmixed into individual channels and saved as a separate file. These files were analysed using AQU Analysis software. DAPI was used to generate a binary mask of all viable cells in the image. Similarly, MLKL, CD47, CitH3 and E-cadherin expression based on the intensity was used in conjunction with DAPI to create binary masks of all cells expressing these biomarkers of interest.

## RNA-Seq

We performed RNA-seq analysis for three independent sample as indicated. For macrophages that co-cultured with PANC-1-OE-Vector or PANC-1-OE-MLKL cells, we performed the RNA-seq in three independent experiments. For wild-type PANC-1 that treated by culture medium derived from co-culture experiments (PANC-1-OE-Vector and macrophages, 1:1, directly co-culture for 72 h) or from co-culture experiments (PANC-1-OE-MLKL and macrophages, 1:1, directly co-culture for 72 h), we performed the RNA-seq in three independent experiments. Total RNA was extracted using Trizol reagent kit (Invitrogen, Carlsbad, CA, USA) according to the manufacturer's protocol. RNA quality was assessed on an Agilent 2100 Bioanalyzer (Agilent Technologies, Palo Alto, CA, USA) and checked using RNase free agarose gel electrophoresis. After total RNA was extracted, eukaryotic mRNA was enriched by Oligo(dT) beads. Then the enriched mRNA was fragmented into short fragments using fragmentation buffer and reversely transcribed into cDNA by using NEBNext Ultra RNA Library Prep Kit for Illumina (NEB #7530, New England Biolabs, Ipswich, MA, USA). The purified double-stranded cDNA fragments were end repaired, A base added, and ligated to Illumina sequencing adapters. The ligation reaction was purified with the AMPure XP Beads (1.0X). And polymerase chain reaction (PCR) amplified. The resulting cDNA library was sequenced using Illumina Novaseq6000 by Gene Denovo Biotechnology Co. (Guangzhou, China). Reads obtained from the sequencing machines includes raw reads containing adapters or low-quality bases which will affect the following assembly and analysis. Thus, to get high quality clean reads, reads were further filtered by fastp (version 0.18.0). The parameters were as follows: (1) removing reads containing adapters; (2) removing reads containing more than 10% of unknown nucleotides(N); (3) removing low quality reads containing more than 50% of low quality (Q-value ≤ 20) bases. The mapped reads of each sample were assembled by using StringTie (v1.3.1) in a reference-based approach. Normalization was performed using DESeq2 pipeline, by normalizing with the size factor. This method is implemented in the R Bioconductor package DESeq2. Differential expression analysis was performed by DESeq pipeline (DESeq2 R package). The cut offs for gene selection were adjusted P < 0.05. Differentially expressed genes (fold change > 1, adjusted P < 0.05) were identified for further analysis. RNA-seq analysis was supported by Gene Denovo Biotechnology Co. Ltd. (Guangzhou, China) and GeneChem Co., Ltd. (Shanghai, China).

## ScRNA-Seq

Primary tumours from C57BL/6 orthotopic model were harvested when four weeks after injection (n = 3 for OE-Vector group; n = 3 for OE-MLKL group), and then dissected and digested by collagenase II, IV (0.25%), and DNase I (0.05%) in DMEM/F12 medium for 30 min at 37 °C. Cells were counted, and $1.5 × 10^4$ primary cancer cells for each tumour were loaded into the 10X Genomics Chromium platform. Samples were processed following the manufacturer's protocol with Single Cell 3′ v2 re agent and then sequenced using an Illumina NextSeq sequencer. Generation of gel beads in emulsion (GEMs), barcoding, GEM-RT clean-up, complementary DNA amplification and library construction were all performed as per the manufacturer's protocol. Qubit was used for library quantification before pooling. The final library pool was sequenced on the Illumina Nova6000 instrument using 150-base-pair paired-end reads. Raw data were mapped to mouse genome mm10-3.0.0 with cellranger 3.1.0. Downstream analyses were performed in R (v.4.0.3). Unsupervised clustering was performed with R (Seurat package version 2.2). Genes expressed in fewer than two cells were filtered out. Cells with >200 genes and <10% mitochondrial genes were further processed. Then, variation coefficient of genes was calculated with Seurat. Dimensionality reduction of data was performed by using principal component analysis based on the first 2000 highest variable genes. A k-nearest neighbour graph was constructed from

Euclidean distances in the space of the first 10 significant principal components. Louvain Modularity optimization algorithm was utilized to cluster the cells in the graph and clustering results were visualized by using t-distributed Stochastic Neighbour Embedding (tSNE) project. Cells expressing high levels of genes encoding haemoglobin were discarded. Differential expression of each cluster was calculated using the 'bimod' test as implemented in Seurat FindMarkers function. Genes with a log2 average expression difference 0.585 and P < 0.05 were identified as marker genes. Cell clusters were annotated using canonical markers of known cell types according to database CellMaker-2.0[82]. Gene ontology enrichment analysis for these significant differentially expressed genes was performed by TopGO R package and the KEGG pathway enrichment analysis was performed using the Hypergeometric test in R. Significantly enriched GO terms and KEGG pathways were selected by a threshold FDR (adjusted P-value) ≤ 0.05. ScRNA-seq was supported by Panomix Biomedical Tech Co., Ltd. (Suzhou, China).

## Statistical analysis

All statistical analyses were performed using GraphPad Prism version 9.1.2. Data are presented as means ± SEMs unless otherwise indicated. Statistical comparisons of continuous variables between two groups were performed using two-sided Student's $t$ test or the Mann-Whitney $U$ test, as appropriate. Categorical variables were analysed using two-sided Pearson's $\chi^2$ test or Fisher's exact test, as appropriate. Comparisons of multiple groups were performed using one-way ANOVA or the Kruskal–Wallis test, as appropriate. Kaplan-Meier survival curves were plotted, and the significance of differences between curves was estimated using the log-rank test. A $p$ value below 0.05 was considered statistically significant.

## Study approval

Human PDAC tissues from Fujian Provincial Hospital were used for IHC, generation of PDOs, and RNA-Seq. PDAC tissue microarray sections from Shanghai Outdo Biotech Company (Shanghai, China) were used for mIFs. The study protocol was approved and authorized by the Ethics Committee of Fujian Provincial Hospital (IRB Number: K2021-03-046, K2023-03-023). All recruited volunteers provided written informed consent. Our study is compliant with the 'Guidance of the Ministry of Science and Technology (MOST) for the Review and Approval of Human Genetic Resources', which requires formal approval for the export of human genetic material or data from China. The patient-related information is available in Supplementary Table 1 and Supplementary Table 2. All animal study procedures and experiments were reviewed and approved by the IACUC of Fujian Medical University (IACUC FJMU 2022-0800) and were in accordance with NIH guidelines for the care and use of animals.

## Reporting summary

Further information on research design is available in the Nature Portfolio Reporting Summary linked to this article.

## Data availability

The scRNA-seq data have been deposited in the GEO database under the GSE266613. The RNA-seq data have been deposited in the GEO database under the GSE248494 and GSE262562. The FACS data have been deposited in the FlowRepository database under the FR-FCM-Z7HD. All data are included in the Supplementary Information or available from the authors, as are unique reagents used in this Article. The raw numbers for charts and graphs are available in the Source Data file whenever possible. Source data are provided with this paper.

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

## Acknowledgements

This work was supported by the Fujian Province Science and Technology Innovation Joint Fund Project (Shi Chen, 2023Y9334), the Major Scientific Research Project of Young and Middle-aged people of Fujian Provincial Health Commission (Shi Chen, 2021ZQNZD001), the Fujian Research and Training Grants for Young and Middle-aged Leaders in Healthcare (Shi Chen, 2021[60]), the National Natural Science Foundation of China (Shi Chen, 82173250), the Special Funding Project of Fujian Provincial Department of Finance (Shi Chen, 2100201), Natural Science Foundation of Fujian Province, Youth Program (Long Huang, 2022J05212), Fujian Provincial Health Science and Technology Plan Project, Youth and Middle-aged Talent Training Project (Long Huang, 2021GGA006). This work was sponsored by the Key Clinical Specialty Discipline Construction Program of Fujian, P.R.C. This work was supported by Geriatric Center Construction Program of Fujian Provincial Medical Creating Double-high Project. We thank the technical supports from Prof. Yi Huang and the colleagues of the Center for Experimental Research in Clinical Medicine, Fujian Provincial Hospital (Fuzhou, China). We also thank the technical supports from Panomix Biomedical Tech Co., Ltd. (Suzhou, China), the Anburui BD Laboratory (Fuzhou, China), Gene Denovo Biotechnology Co., Ltd. (Guangzhou, China), Shanghai Outdo Biotech Company (Shanghai, China), GeneChem Co., Ltd. (Shanghai, China), Fuzhou Sunya Biotechnology Co., Ltd. (Fuzhou, China), Servicebio (Wuhan, China), AimingMed Technologies Co., Ltd. (Hangzhou, China) and Keystone Biotechnology Co., Ltd. (Fuzhou China). Figure 1a, Fig. 3d, Fig. 4k, Fig. 7a, Fig. 8a, Supplementary Fig. 1g, Supplementary Fig. 4i, Supplementary Fig. 6a, Supplementary Fig. 6e, Supplementary Fig. 7a, Supplementary Fig. 7e, Supplementary Fig. 8e, Supplementary Fig. 12a, Supplementary Fig. 12b, Supplementary Fig. 13 Created with BioRender.com released under a Creative Commons Attribution-NonCommercial-NoDerivs 4.0 International license (https://creativecommons.org/licenses/by-nc-nd/4.0/deed.en).

## Author contributions

Shi Chen and Cheng-Yu Liao designed the study. Cheng-Yu Liao, Ge Li, Feng-Ping Kang performed the study and wrote the paper. Shi Chen, Zu-Wei Wang and Yi-Feng Tian conducted the experiments. Yong-Ding Wu, Cheng-Ke Xie and Hong-Yi Lin collected the tissues. Cai-Feng Lin participated in data analysis. Jian-Fei Hu, Xiao-Xiao Huang, Shun-Cang Zhu, Long Huang and Jian-Lin Lai assisted with experiments. Yi-Feng Tian, Yi Huang, Qiao-Wei Li and Li-Qun Chen provided technical help.

## Competing interests

The authors declare no competing interests.
