## [Peer Review File · Nature Communications]

REVIEWER COMMENTS

Reviewer #1 (MET)(Remarks to the Author):

This is a spectacular study that uses state-of-the-art technology to elucidate the detailed steps of the mechanism of liver metastasis in early T-stage of PDAC. I recognize that the paper consists of the following key steps and cannot point out any obvious logical leaps.

1. The authors demonstrated that necroptosis driven by MLKL correlates with liver metastasis in early-T stage PDAC using biopsy tissue and organoid models of early-T stage PDAC patients with liver metastasis as material.
2. They showed that MLKL-driven cancer cell necroptosis interacts with macrophages to induce macrophage infiltration and upregulation of cancer cell CD47, which is responsible for cancer cell resistance to phagocytosis.
3. The authors demonstrated that this phenomenon leads to the formation of macrophage extracellular traps (METs) and successfully identified CXCL-8 as a METs inducer from culture supernatants of PDAC cancer cells. Furthermore, METs have been shown to play an important role in multiple metastatic stages by trapping and activating CXCL-8.
4. Successfully demonstrated that the combination of anti-CD47 and MLKL inhibitor prevents liver metastasis of PDAC and ultimately prolongs survival, which is of great clinical significance as a promising neoadjuvant combination therapy for early stage T PDAC
5. In the future, the combination of these therapies with synergistic anticancer drugs may be one of the major pathways to complete cure of PDAC.

Reviewer #2 (Necroptosis mediated immune regulation)(Remarks to the Author):

Liao and colleagues detect increased MLKL expression in PDAC metastatic samples (T1M1 vs T1M0) and then examine the effect(s) of MLKL overexpression (MLKL O/E) in human pancreatic cell lines in vitro and in vivo in syngeneic and immune deficient disease models and using organoids. They show that MLKL O/E stimulates MLKL phosphorylation and oligomerization resulting in necroptosis that is independent of RIPK1 and RIPK3. The cell death stimulates macrophage infiltration, increased expression of the “Do not eat me” signal CD47 and MET formation. MET formation localizes CXCL8 expression for MMP12 cleavage of CXCL8. There is a substantial amount of data supporting their model of PDAC metastasis however the authors rely on descriptive measures that are not quantitative.

Although this area is of high interest to the field, the key question as to why MLKL is O/E in PDAC mets remains unaddressed. We know from several reports in the literature that MLKL O/E results in MLKL oligomerization and activation and induction of necroptosis. In a co-culture or in vivo setting this is expected to result in DAMP release and innate immune activation, cytokine and chemokine production. The critical question is what triggers MLKL upregulation in PDAC progression and whether MLKL inhibition can prevent PDAC metastasis.

Specific questions: Does Ras and/or other genetic events such as p53 loss trigger MLKL O/E or is this a response to the microenvironment? What is the effect of MLKL inhibition or MLKL loss on PDAC initiation

vs progression? Does macrophage removal in B6 mice prevent/reduce liver metastasis or do other cell types then substitute? Is it necessary to invoke MET formation as it seems as if the macrophage necroptosis could result in cytokine/chemokine and DNA release? Do pancreatic ductal epithelial cells express Ripk1, Ripk3 and Mkl and are they sensitive to TNF or IFNg induced necroptosis? Or do most PDAC tumors resemble PANC-1 with no detectable RIPK3?

Minor comments:

1. What does aneuploid vs diploid refer to in Fig 3E? Are there more M1 macrophages detected by scRNAseq in T1M1 tumors?
2. Several statements are made in the ms but not referenced. For ex, "These cytokines (CXCL-8, IL-6, GM-CSF) are well-documented for promoting tumor metastasis by enhancing tumor cell EMT through their receptors (CXCR1/2, IL6R, GM-CSFR)." No refs
3. Trajectory inference implicates CXCL9 macrophages what about CXCL8?
4. P.11 line 2 should one of these be M1?
5. Please show controls for CC3 and CC8 WB (Fig 1M).
6. It was difficult to discern when effects of MLKL O/E were examined in spontaneous metastasis models opposed to lung/liver colonization assays achieved via IV injection. How are the liver mets detected and quantified? Or are they liver colonies? It is helpful to distinguish what these assays measure.
7. The manuscript contains many typos and grammatical errors and could benefit from additional editing. The multi-panel figures are too small and are difficult to read even when enlarged/magnified.

Reviewer #3 (Carcinoma)(Remarks to the Author):

Interesting and remarkable paper. I enjoyed reading this article reporting the impressive "Don't eat me, Let's Run" metastasis strategy of pancreatic cancer, bringing new insights into tumor metastasis and necroptosis. To the best of my knowledge, this is the first detailed investigation into the macrophage extracellular traps of macrophages in the field of cancer. The authors provided adequate evidences to demonstrate the role of macrophage extracellular traps in metastasis steps and the mechanism of CXCL8 activation that cleaved by macrophage extracellular traps. Generally, these greatly helps to gain a more comprehensive understanding on the role of macrophages in cancer field and the different functional modules of macrophages. The T1M1-PDAC, live metastasis in an early stage as the authors defined, is a pathetic clinical dilemma for surgeon and patients. And this research proposed "GW+anti-CD47" provides an opportunity for the radical surgery. I found that the article was generally well written and recommended an acceptance for publication in Nature Communications. Here, I have only some minor comments:

1. As described by the author, MLKL-driven necroptosis brings a series of immune events represented by macrophages. Why not consider direct single-cell sequencing in patients with T1M0-PDAC and T1M1-PDAC?
2. Are the samples for constructing organoids (Fig 1K) derived from patients undergoing RNA-seq and MRI (Fig 1B-1C)? If so, please provide a clear description.
3. The author showed increased macrophages infiltration in the OE-MLKL group of liver metastasis model (Fig 2F), I suggested the authors to show the macrophages infiltration in orthotopic model.
4. Page 11 Line 9, the upregulation of CD86 and CD163 were derived from RNA-seq (Fig 3L), it should be further validated in vitro.
5. Further identification of which subtype of macrophages produced METs may seem interesting, but the author did not analyze it. If it is due to limitations in the entire research content, I suggest discussing this issue in the discussion section.
6. The specific phosphorylation sites of p-MLKL, p-RIPK1 and p-RIPK3 should be indicated. After all, only certain specific phosphorylation sites represent the level of necroptosis.
7. The western blot of N-cadherin in ASPC1 cells and the blots of RIPK3/p-RIPK3 were nearly invisible, please confirm.
8. Part of the font in the figures is small, please check all figures.
9. All scale bars are small and should be clearly displayed.
10. Primer, as well as antibody related information, should be provided.

REVIEWER COMMENTS

Reviewer #1 (MET)(Remarks to the Author):

This is a spectacular study that uses state-of-the-art technology to elucidate the detailed steps of the mechanism of liver metastasis in early T-stage of PDAC. I recognize that the paper consists of the following key steps and cannot point out any obvious logical leaps.

1. The authors demonstrated that necroptosis driven by MLKL correlates with liver metastasis in early-T stage PDAC using biopsy tissue and organoid models of early-T stage PDAC patients with liver metastasis as material.
2. They showed that MLKL-driven cancer cell necroptosis interacts with macrophages to induce macrophage infiltration and upregulation of cancer cell CD47, which is responsible for cancer cell resistance to phagocytosis.
3. The authors demonstrated that this phenomenon leads to the formation of macrophage extracellular traps (METs) and successfully identified CXCL-8 as a METs inducer from culture supernatants of PDAC cancer cells. Furthermore, METs have been shown to play an important role in multiple metastatic stages by trapping and activating CXCL-8.
4. Successfully demonstrated that the combination of anti-CD47 and MLKL inhibitor prevents liver metastasis of PDAC and ultimately prolongs survival, which is of great clinical significance as a promising neoadjuvant combination therapy for early stage T PDAC
5. In the future, the combination of these therapies with synergistic anticancer drugs may be one of the major pathways to complete cure of PDAC.

Response: Thanks a lot. We are very excited for your recognition of our work.

Reviewer #2 (Necroptosis mediated immune regulation)(Remarks to the Author):

Liao and colleagues detect increased MLKL expression in PDAC metastatic samples (T1M1 vs T1M0) and then examine the effect(s) of MLKL overexpression (MLKL O/E) in human pancreatic cell lines in vitro and in vivo in syngeneic and immune deficient disease models and using organoids. They show that MLKL O/E stimulates MLKL phosphorylation and oligomerization resulting in necroptosis that is independent of RIPK1 and RIPK3. The cell death stimulates macrophage infiltration, increased expression of the "Do not eat me" signal CD47 and MET formation. MET formation localizes CXCL8 expression for MMP12 cleavage of CXCL8. There is a substantial amount of data supporting their model of PDAC metastasis however the authors rely on descriptive measures that are not quantitative.

Although this area is of high interest to the field, the key question as to why MLKL is O/E in PDAC mets remains unaddressed. We know from several reports in the literature that MLKL O/E results in MLKL oligomerization and activation and induction of necroptosis. In a co-culture or in vivo setting this is expected to result in DAMP release and innate immune activation, cytokine and chemokine production. The critical question is what triggers MLKL upregulation in PDAC progression and whether MLKL inhibition can prevent PDAC metastasis.

Response: Thanks for your professional and thoughtful comments to improve the manuscript. We have provided point-by-point responses to your concerns.

Specific questions:

1. Does Ras and/or other genetic events such as p53 loss trigger MLKL O/E or is this a response to the microenvironment?

Response: Thank you for this interesting query. According to your suggestion, we further explored this aspect but could not identify a direct link between Ras/TP53 and MLKL O/E.

(1) Prior research has identified KRAS and TP53 as the primary driver genes responsible for genomic events in PDAC ^[1]. We checked our RNA-seq data and found no significant difference in KRAS/TP53 mutations between the T1M1 group with aberrant elevated MLKL expression and the T1M0 group (KRAS mutation: 4/6 in T1M0 vs. 3/6 in T1M1, p=0.558; TP53 mutation: 5/6 in T1M0 vs. 5/6 in T1M1, p=1.000). In order to further investigate the relationship between MLKL expression and KRAS mutations, we conducted a thorough analysis of PDAC cell lines with and without KRAS mutations, using the external CCLE database. Our analysis revealed a modest increase in MLKL expression in PDAC cell lines with KRAS mutation. However, this increase was not statistically significant. Specifically, we classified KRAS mutations into G12C, G12D, G12V, and other KRAS mutations, and no significant associations between MLKL expression and different KRAS-mutation types were found (**as below**).

Figure R1. MLKL expression of CCLE PDAC cell lines with different KRAS and TP53 mutations.

(2) Furthermore, we investigated the impact of KRAS mutation on MLKL expression *in vitro* (as below) through HEK293T cells that were transfected with wild-type/mutated KRAS (G12C/G12D/G12V). The results did not reveal a significant association between them.

Figure R2. MLKL RNA and protein expression of HEK293T transfected with wild-type KRAS and different KRAS mutations.

(3) We followed your suggestion and investigated whether overexpression of MLKL is a response to TME. We implanted PA02 cells (without KRAS mutation) orthotopically in NOD-SCID and C57BL/6 mice (as indicated below), then collected the tumor cells to examine the expression of MLKL after euthanasia. We found that MLKL was not significantly upregulated in C57BL/6 mice when compared to those cultured *in vitro* and implanted in NOD-SCID mice.

Figure R3. MLKL expression of different treated PA02 cells.

Based on these investigations, the overexpression of MLKL doesn't appear to be driven by KRAS/TP53 events or a response to TME. We observed elevated levels of MLKL at both RNA and protein levels in T1M1. Subsequently, we analyzed the other possible reasons for this increase, which are summarized below:

- a) Increased transcription: We analyzed the TFs of MLKL predicted by JASPAR database preliminarily, and they were not increased in T1M1 PDAC.

Figure R4. Predicted transcriptional factors of MLKL according to JASPAR.

- b) Previously-reported regulatory factors for MLKL expression, such as non-coding RNA regulations: LncRNA-FA2H-2, LncRNA HABON [2-4], CREB (CRTC2) were not found to be changed in T1M1-PDAC based on our RNA-seq data.

Figure R5. Predicted transcriptional factors and other previously-reported factors that regulating MLKL expression in RNA-seq data.

- c) Post-transcriptional regulations: we first tested the pre-mRNA of MLKL in T1M1-PDOs and T1M0-PDOs, and no significant difference was observed. However, the elevated RNA level of MLKL caught our attention, indicating an abnormal alternative splicing process on MLKL, which may ultimately result in the overexpression of MLKL in T1M1-PDAC [5].

Figure R6. MLKL pre-mRNA of T1M0-PDO and T1M1-PDO.

We have added these results and discussion to the revised manuscript (**Page 8 line 3-6, Page 21 line 10-13**).

2. What is the effect of MLKL inhibition or MLKL loss on PDAC initiation vs progression?

Response: Following your suggestion, we examined MLKL-knockout cells for the impacts of MLKL loss on PDAC initiation and progression [6]. The absence of MLKL inhibited PDAC cells' colony formation and 3D tumor spheroids formation, and it also reduced their invasive and migratory capacities (**as below A-D**). A decrease in tumor burden and liver metastasis were found in mice transplanted with MLKL-knockout cells, both in the orthotopic and liver metastasis tumor sites (**as below E-F**). Therefore, we conclude that MLKL loss could inhibit PDAC initiation and progression. Accordingly, we added this content to the

Results and Discussion section in the revised manuscript (Page 10, Line 2-3; Page 21 Line 13-18).

Figure R7. (A) 2D tumor colony formation of PDAC cells with control/MLKL-knockout; (B) 3D tumor spheroid formation of PDAC cells with control/MLKL-knockout; (C-D) 3D tumor spheroid invasion and migration of PDAC cells with control/MLKL-knockout; (E) C57BL/6 orthotopic and liver metastasis model that injected with KPC cells with control/MLKL-knockout.

While MLKL loss of function seems interesting in PDAC progression, our interest lies in MLKL-driven necroptosis' role in PDAC metastasis. This is why we mainly used MLKL-overexpression models in our study. In addition, it has been recently reported that MLKL plays a role in other process besides necroptosis, including receptor internalization, ligand-receptor degradation, endosomal trafficking, extracellular vesicle formation, autophagy, nuclear functions, axon repair, endoplasmic reticulum regulation, and inflammasome regulation [7-8]. We were concerned that the inhibition or loss of MLKL in the perturbations introduces confounding factors for the delineation of MLKL-driven necroptosis process.

- Does macrophage removal in B6 mice prevent/reduce liver metastasis or do other cell types then substitute?

Response: Yes, macrophage depletion reduced the liver metastasis in our study (from 87.5% to 25%), as indicated in Fig 2G, Fig 2J., which highlighted the vital role of macrophages in the context. However, the depletion could not completely prevent the liver metastasis event, which indicates that there might be other substituting cell types.

After a careful check and analysis of our existing data, no clear cell types emerge. We have added this interesting discussion to the revised manuscript (**Page 22 Line8-15**).

4. Is it necessary to invoke MET formation as it seems as if the macrophage necroptosis could result in cytokine/chemokine and DNA release?

Response: We agree that distinguishing MET formation from macrophage necroptosis is crucial to supporting our findings. When exposed to certain stimuli, inhibition of necroptosis can restore the formation of METs in macrophages, indicating that METs and necroptosis can be transformed into each other [9]. We further examined the necroptosis level of macrophages co-cultured with Vector/MLKL overexpressing PADC cells (**as below**). However, the results did not suggest a difference in the necroptosis level of macrophages.

Some studies reported that METs form during METosis [10], and macrophages that produce METs will no longer survive [11-13]. Different from necroptosis, the nuclear and granular membranes collapse while the plasma membrane is intact in this ETosis [14-16]. Therefore, in this study we used the non-cellular permeable SYTOX-orange to stain the extracellular DNA, further clarifying the survival status of macrophages. This also indicates that macrophages did not go through extensive necroptosis in our experiments. We have added this part in the Results section and Discussion section of the revised manuscript (**Page 14 Line19-21, Page 22 Line16-19**).

Figure R8. WB analysis of necroptosis level of macrophages that co-cultured with PDAC cells with Vector/MLKL overexpression for 72h.

5. Do pancreatic ductal epithelial cells express Ripk1, Ripk3 and Mlkl and are they sensitive to TNF or IFNg induced necroptosis? Or do most PDAC tumors resemble PANC-1 with no detectable RIPK3?

Response: We believe that you may be seeking clarification on whether the difference in MLKL in T1M1 and T1M0 tumor cells is due to inherent variance in pancreatic epithelial cells. Our analysis of the tumor and its adjacent tissue revealed no difference in MLKL and RIPK1/3 expression between adjacent normal tissues. This supports that the high expression of MLKL in T1M1 is directly associated with the existence of metastasis, rather than other variances.

Following your suggestion, we examined RIPK1, RIPK3, and MLKL expression in pancreatic ductal epithelial cell-line (HPNE) through WB and in pancreatic duct tissue from normal human pancreas tissue through IHC (**figure A-E below**). The expression levels of these proteins in normal tissue or cell lines were significantly lower than those in tumor cells or tumor tissue (**figure A below**), which was consistent with the external GEPIA2 database (**figure C below**). In addition, we also analyzed the expression of RIPK1, RIPK3, and MLKL in the para-cancerous tissues of T1M0 and T1M1 based on our RNA-seq data (**figure B below**), which revealed no significant difference.

Although normal cells can also be induced by TNF and ITF, we believe that normal cells still undergo classic necroptosis mode, which is different from the MLKL-driven mode in T1M1 tumors. The pancreatic ductal epithelial cell (HPNE) could be induced into necroptosis by TNF or IFN γ (**figure D below**), and it could be rescued by RIPK3 inhibitor (GSK-872) (**figure E below**). This is different from the MLKL-driven necroptosis in T1M1 tumor cells that could not be rescued by RIPK3 inhibitor in our study. This further supported the specificity of MLKL-driven necroptosis in tumor cells rather than pancreatic ductal epithelial cells.

In addition, we also evaluated the expression of MLKL in different stages of PDAC and found that the differences among AJCC stage I-IV were not significant in the large sample data (GEPIA2 database) (**figure F below**), but it was relatively lowest in AJCC Stage I, which is consistent with our low expression of MLKL in T1M0 (AJCC Stage I). And we have added the relative content in the revised manuscript (**Page 8 Line8-11, Page 9 Line13-15, Page 21 Line10-13**).

Figure R9. (A) Representative IHC staining of RIPK1, RIPK3, and MLKL in T1M0 and T1M1 PDAC tissue and normal tissue; (B) The RIPK1, RIPK3, and MLKL expression heatmap in tumor tissue and normal tissue of T1M0 and T1M1 PDAC according to the RNA-seq data.(C) The RIPK1, RIPK3 and MLKL expression in GEPIA2 database;(D-E) WB analysis of HPNE necroptosis induced by TNF, IFN γ and it could be rescued by RIPK3 inhibitor(GSK-872), as well as RIPK1 inhibitor (Nec-1) and MLKL inhibitor (GW); (E) The MLKL expression in different PDAC AJCC-stages according to GEPIA2 database.

Regarding the expression of RIPK3, we apologized for our unclear descriptions. PDAC samples express RIPK3, but the levels vary a lot, as shown in different PDAC cell lines and TCGA-PDAC samples (as below).

We used two cell lines, AsPC1 (RIPK3-high) and PANC1 (no detectable RIPK3), to simulate the aberrant elevated MLKL-driven necroptosis of T1M1-PDAC. Regardless of the presence or absence of RIPK3, both of them showed consistent MLKL-driven necroptosis and subsequent TME events. That highlighted the unique and pivotal role of MLKL that massive MLKL has strong toxicity to cause necroptosis, which is consistent with previous reports [17-19]; And we have updated a clear description in the revised manuscript (**Page 9 Line3-5, Page 21 Line 13-18**).

Figure R10. (A)RIPK1, RIPK3 and MLKL expression of CCLC PDAC cell lines; (B) The RIPK1, RIPK3 and MLKL expression of TCGA PDAC data;

Minor comments:

1. What does aneuploid vs diploid refer to in Fig 3E? Are there more M1 macrophages detected by scRNAseq in T1M1 tumors?

Response: Here, we use CNV characteristics to classify epithelial cells into cancer cells and pancreatic ductal cells. Aneuploid means cancer cells and diploid means pancreatic ductal cells. We have revised the figure legend of **Fig 3e** of the revised manuscript.

Yes, there are more M1 macrophages detected by scRNAseq *in vivo* (**Fig 3h**), and the M1 macrophage score in T1M1 PDAC appears to be higher when analyzed by RNA-seq data (**Fig 3j**). We make more clear descriptions about that in the revised manuscript (**Page 11 line 5-9, Page 11 Line16-18**).

2. Several statements are made in the ms but not referenced. For ex, "These cytokines (CXCL-8, IL-6, GM-CSF) are well-documented for promoting tumor metastasis by enhancing tumor cell EMT through their receptors (CXCR1/2, IL6R, GM-CSFR)." No refs

Response: Thanks for your careful comments, we added these citations (as below) in the revised manuscript.

[1] Liu Q, Li A, Tian Y, et al. The CXCL8-CXCR1/2 pathways in cancer. *Cytokine Growth Factor Rev.* 2016;31:61-71.

[2] Liu F, Liang Y, Sun R, et al. Astragalus mongholicus Bunge and Curcuma aromatica Salisb. inhibits liver metastasis of colon cancer by regulating EMT via the CXCL8/CXCR2 axis and PI3K/AKT/mTOR signaling pathway. *Chin Med.* 2022;17(1):91.

[3] Rokavec M, Öner MG, Li H, et al. IL-6R/STAT3/miR-34a feedback loop promotes EMT-mediated colorectal cancer invasion and metastasis. *J Clin Invest.* 2014;124(4):1853-1867.

[4] Chen Y, Zhao Z, Chen Y, et al. An epithelial-to-mesenchymal transition-inducing potential of granulocyte macrophage colony-stimulating factor in colon cancer. *Sci Rep.* 2017;7(1):8265.

[5] Hong C, Schubert M, Tijhuis AE, et al. cGAS-STING drives the IL-6-dependent survival of chromosomally unstable cancers. *Nature.* 2022;607(7918):366-373.

[6] Klemm F, Möckl A, Salamero-Boix A, et al. Compensatory CSF2-driven macrophage activation promotes adaptive resistance to CSF1R inhibition in breast-to-brain metastasis. *Nat Cancer.* 2021;2(10):1086-1101.

[7] Fousek K, Horn LA, Palena C. Interleukin-8: A chemokine at the intersection of cancer plasticity, angiogenesis, and immune suppression. *Pharmacol Ther.* 2021;219:107692.

[8] Gulhati P, Schalck A, Jiang S, et al. Targeting T cell checkpoints 41BB and LAG3 and myeloid cell CXCR1/CXCR2 results in antitumor immunity and durable response in pancreatic cancer. *Nat Cancer.* 2023;4(1):62-80.

[9] Liu H, Zhao Q, Tan L, et al. Neutralizing IL-8 potentiates immune checkpoint blockade efficacy for glioma. *Cancer Cell.* 2023;41(4):693-710.e8.

[10] Lopez-Bujanda ZA, Haffner MC, Chaimowitz MG, et al. Castration-mediated IL-8 promotes myeloid infiltration and prostate cancer progression. *Nat Cancer.* 2021;2(8):803-818.

3. Trajectory inference implicates CXCL9 macrophages what about CXCL8?

Response: Here, we defined the macrophage clusters with high CXCL9 expression as CXCL9+ macrophages through scRNA-seq according to the previous study [20]. Macrophage clusters with CXCL8 high expression were not found in our single-cell data.

4. P.11 line 2 should one of these be M1?

Response: We have corrected the error; thank you (**Page 11, Line 13**).

5. Please show controls for CC3 and CC8 WB (Fig 1M).

Response: We have added the corresponding controls for CC3 and CC8 in the revised manuscript (**Fig 1m, as below**).

Figure R11. WB analysis of apoptosis level of PDAC cells with Vector/MLKL overexpression.

6. It was difficult to discern when effects of MLKL O/E were examined in spontaneous metastasis models opposed to lung/liver colonization assays achieved via IV injection. How are the liver mets detected and quantified? Or are they liver colonies? It is helpful to distinguish what these assays measure.

Response: It was challenging to quantify liver metastasis compared to the well-distinguished lung/liver colonization assays. Therefore, we employed the incidence rate of liver metastasis as a quantification approach based on the following failed attempts:

- (1) In our study, we attempted to measure liver metastasis using luciferase-based bioluminescence imaging. However, we encountered difficulty in accurately quantifying the tumor burden of liver metastasis and the primary pancreatic tumor separately, which was necessary for our study. The fluorescence of the spontaneous metastasis model is often fused together, resulting in an overall measurement of the tumor burden, as indicated in **Fig 8l**.
- (2) We then attempted to use micro-CT to evaluate the liver metastasis before sacrifice, as indicated by irregular patchy low-density lesions of micro-CT (**Fig 2i**). However, it's challenging to apply consistent criteria for the measurements. Considering that PDAC is a tumor with poor blood supply, we did not use enhanced CT for evaluation here.
- (3) From the perspective of specimens, liver metastases are often connected in a patchy manner, with some cases combining multiple lesions/colonies, which is not an obvious liver colony (**as indicated below, source figure of Figure 2i**). This form is consistent with some previous literature ^[21-22]. This is also difficult to quantify by counting colony lesions. Based on all these considerations, we finally decided to use the incidence rate of liver metastasis to evaluate the higher risk of liver metastasis of the MLKL group of the orthotopic model.

Figure R12. Specimen from C57BL/6 mice orthotopically injected with Vector/MLKL overexpression; white circles and white arrows indicate liver-metastasis.

7. The manuscript contains many typos and grammatical errors and could benefit from additional editing. The multi-panel figures are too small and are difficult to read even when enlarged/magnified.

Response: Based on your helpful comments, we have further polished the language of our manuscript through Springer Nature language service and re-edited the figures for easy reading.

SPRINGER NATURE **Editing Certificate**

Author Services

This document certifies that the manuscript

Necroptosis enhances “Don’t eat me” signal and induces macrophage extracellular traps to promote pancreatic cancer liver metastasis

prepared by the authors

Chengyu Liao

was edited for proper English language, grammar, punctuation, spelling, and overall style by one or more of the highly qualified native English speaking editors at SNAS.

This certificate was issued on **January 26, 2024** and may be verified on the SNAS website using the verification code **7D53-C408-3B5A-F1FB-957P**.

Neither the research content nor the authors' intentions were altered in any way during the editing process. Documents receiving this certification should be English-ready for publication; however, the author has the ability to accept or reject our suggestions and changes. To verify the final SNAS edited version, please visit our verification page at secure.authorservices.springernature.com/certificate/verify.
If you have any questions or concerns about this edited document, please contact SNAS at support@as.springernature.com.

SNAS provides a range of editing, translation, and manuscript services for researchers and publishers around the world. For more information about our company, services, and partner discounts, please visit authorservices.springernature.com.

Reviewer #3 (Carcinoma)(Remarks to the Author):

Interesting and remarkable paper. I enjoyed reading this article reporting the impressive "Don't eat me, Let's Run" metastasis strategy of pancreatic cancer, bringing new insights into tumor metastasis and necroptosis. To the best of my knowledge, this is the first detailed investigation into the macrophage extracellular traps of macrophages in the field of cancer. The authors provided adequate evidences to demonstrate the role of macrophage extracellular traps in metastasis steps and the mechanism of CXCL8 activation that cleaved by macrophage extracellular traps. Generally, these greatly helps to gain a more comprehensive understanding on the role of macrophages in cancer field and the different functional modules of macrophages. The T1M1-PDAC, live metastasis in an early stage as the authors defined, is a pathetic clinical dilemma for surgeon and patients. And this research proposed "GW+anti-CD47" provides an opportunity for the radical surgery. I found that the article was generally well written and recommended an acceptance for publication in Nature Communications. Here, I have only some minor comments:

Response: we are grateful and excited for your comments on our study.

1.As described by the author, MLKL-driven necroptosis brings a series of immune events represented by macrophages. Why not consider direct single-cell sequencing in patients with T1M0-PDAC and T1M1-PDAC?

Response: We appreciate this thoughtful comment. We had originally designed to use scRNA-seq on T1M1-T1M0 to profile the TME landscape. The biopsy samples from T1M1-PDAC were limited by tissue amount and necrosis status, which led to inconclusive single-cell analysis and could not reflect the TME landscape well.

2.Are the samples for constructing organoids (Fig 1K) derived from patients undergoing RNA-seq and MRI (Fig 1B-1C)? If so, please provide a clear description.

Response: Yes, the samples for constructing organoids were derived from patients undergoing RNA-seq and MRI in our study. According to your suggestion, we added a clear description in the revised manuscript (**Page 8, Line 20-22**) and the corresponding figure legend (**Page 47, legend for Fig 1k**).

3.The author showed increased macrophages infiltration in the OE-MLKL group of liver metastasis model (Fig 2F), I suggested the authors to show the macrophages infiltration in orthotopic model.

Response: Following your suggestions, we have added it in the revised manuscript (**Fig S2h, as below**).

Figure R13. Representative images of immunohistochemistry of macrophage maker F4/80 in primary tumor lesion of C57BL/6 orthotopic model. The triangles indicate the F4/80 positive macrophages. Five mice were in each group. Scale bar: 50 μ m.

4. Page 11 Line 9, the upregulation of CD86 and CD163 were derived from RNA-seq (Fig 3L), it should be further validated in vitro.

Response: Following your suggestions, we have added it to the revised manuscript (**Fig S3h, as below**).

Figure R14. The RNA level of CD86 and CD163 of THP-1 derived macrophage that co-cultured with tumor cell with Vector/MLKL overexpression.

5. Further identification of which subtype of macrophages produced METs may seem interesting, but the author did not analyze it. If it is due to limitations in the entire research content, I suggest discussing this issue in the discussion section.

Response: We have added the relevant content to the Discussion sections of the revised manuscript (**Page 24 Line 1-2**).

6. The specific phosphorylation sites of p-MLKL, p-RIPK1 and p-RIPK3 should be indicated. After all, only certain specific phosphorylation sites represent the level of necroptosis.

Response: We have indicated this in the revised manuscript's corresponding WB blots (**Fig 1j, 1m, 1o**).

7. The western blot of N-cadherin in ASPC1 cells and the blots of RIPK3/p-RIPK3 were nearly invisible, please confirm.

Response: Yes. We have confirmed the extremely low expression of N-cadherin in AsPC-1 cells for several attempts. In qPCR assay, the CT values of N-cadherin in AsPC-1 cells were often higher than 35/undetected. By using the N-cadherin-negative MCF-7 cells (as indicated in the instructions from CST and Proteintech) for negative control and the PANC-1 cells for positive control, we confirmed it again by using two different antibodies (CST#13116, Proteintech# 20874-1-AP; **below figure A**). These results were consistent with the report from Birgit Hotz et al. [23].

Regarding the RIPK3/p-RIPK3, we also have confirmed the extremely no expression of RIPK3 in PANC-1 through qPCR (CT value > 35) and WB by using two different antibodies (CST, Proteintech; positive control: AsPC-1 cells, **below figure B**). And these results were consistent with previous reports [24-25]. We chose the RIPK3-null cells to overexpress MLKL to imitate the RIPK3-independent MLKL-driven necroptosis and similar results were also observed in AsPC-1 cells (with relatively high expression of RIPK3). It means that MLKL-driven necroptosis could happen without or with the existence of RIPK3, which is similar to the view about starvation-induced MLKL phosphorylation independently of RIPK3 [26].

Figure R15. (A) N-cadherin protein level of AsPC-1 cells indicated by two antibodies; (B) RIPK3 protein level of PANC-1 cells indicated by two antibodies.

8. Part of the font in the figures is small, please check all figures.

Response: We have checked and re-generated all the figures according to the figure guideline of *Nature Communications*.

9. All scale bars are small and should be clearly displayed.

Response: We have checked and re-generated all the scale bars in the revised figures.

10. Primer, as well as antibody related information, should be provided.

Response: We provided the information in the Table S3, Source-Data and Report Summary files according to the requirement of *Nature Communications*.

Reference

- [1] Park W, Chawla A, O'Reilly EM. Pancreatic Cancer: A Review. *JAMA*. 2021;326(9):851-862.
- [2] Guida N, Laudati G, Serani A, et al. The neurotoxicant PCB-95 by increasing the neuronal transcriptional repressor REST down-regulates caspase-8 and increases Ripk1, Ripk3 and MLKL expression determining necroptotic neuronal death. *Biochem Pharmacol*. 2017;142:229-241.
- [3] Guo FX, Wu Q, Li P, et al. The role of the LncRNA-FA2H-2-MLKL pathway in atherosclerosis by regulation of autophagy flux and inflammation through mTOR-dependent signaling. *Cell Death Differ*. 2019;26(9):1670-1687.
- [4] Wo L, Zhang X, Ma C, et al. LncRNA HABON promoted liver cancer cells survival under hypoxia by inhibiting mPTP opening. *Cell Death Discov*. 2022;8(1):171.
- [5] Kong Y, Luo Y, Zheng S, et al. Mutant KRAS Mediates circARFGEF2 Biogenesis to Promote Lymphatic Metastasis of Pancreatic Ductal Adenocarcinoma. *Cancer Res*. 2023;83(18):3077-3094.
- [6] Wu C, Rakhshandehroo T, Wettersten HI, et al. Pancreatic cancer cells upregulate LPAR4 in response to isolation stress to promote an ECM-enriched niche and support tumour initiation. *Nat Cell Biol*. 2023;25(2):309-322.
- [7] Martens S, Bridelance J, Roelandt R, Vandenabeele P, Takahashi N. MLKL in cancer: more than a necroptosis regulator. *Cell Death Differ*. 2021;28(6):1757-1772.
- [8] Jiang X, Deng W, Tao S, et al. A RIPK3-independent role of MLKL in suppressing parthanatos promotes immune evasion in hepatocellular carcinoma. *Cell Discov*. 2023;9(1):7.
- [9] Cui Y, Xiao Q, Zhang Q, et al. Black carbon nanoparticles activate the crosstalk mechanism between necroptosis and macrophage extracellular traps to change macrophages fate. *Environ Res*. 2023;232:116321.
- [10] Doster RS, Rogers LM, Gaddy JA, Aronoff DM. Macrophage Extracellular Traps: A Scoping Review. *J Innate Immun*. 2018;10(1):3-13.
- [11] Chow OA, von Köckritz-Blickwede M, Bright AT, Hensler ME, Zinkernagel AS, Cogen AL, Gallo RL, Monestier M, Wang Y, Glass CK, Nizet V: Statins enhance formation of phagocyte extracellular traps. *Cell Host Microbe* 2010;8:445-454.
- [12] Nakazawa D, Shida H, Kusunoki Y, Miyoshi A, Nishio S, Tomaru U, Tsumi T, Ishizu A: The responses of macrophages in interaction with neutrophils that undergo NETosis. *J Autoimmun* 2016;67:19-28.
- [13] Vega VL, Crotty Alexander LE, Charles W, Hwang JH, Nizet V, De Maio A: Activation of the stress response in macrophages alters the M1/M2 balance by enhancing bacterial killing and IL-10 expression. *J Mol Med (Berl)* 2014; 92:1305-1317.
- [14] Fuchs TA, Abed U, Goosmann C, Hurwitz R, Schulze I, Wahn V, Weinrauch Y, Brinkmann V, Zychlinsky A: Novel cell death program leads to neutrophil extracellular traps. *J Cell Biol* 2007;176:231-241.
- [15] Liu P, Wu X, Liao C, et al. Escherichia coli and Candida albicans induced macrophage extracellular trap-like structures with limited microbicidal activity. *PLoS One*. 2014;9(2):e90042.
- [16] Remijsen Q, Kuijpers TW, Wirawan E, Lippens S, Vandenabeele P, Vanden Berghe T. Dying for a cause: NETosis, mechanisms behind an antimicrobial cell death modality. *Cell Death Differ*. 2011;18(4):581-588.
- [17] Hildebrand JM, Tanzer MC, Lucet IS, et al. Activation of the pseudokinase MLKL unleashes the four-helix bundle domain to induce membrane localization and necroptotic cell death. *Proc Natl Acad Sci U S A*. 2014;111(42):15072-15077.
- [18] Dondelinger Y, Declercq W, Montessuit S, et al. MLKL compromises plasma membrane integrity by binding to phosphatidylinositol phosphates. *Cell Rep*. 2014;7(4):971-981.
- [19] Zhan Q, Jeon J, Li Y, et al. CAMK2/CaMKII activates MLKL in short-term starvation to facilitate autophagic flux. *Autophagy*. 2022;18(4):726-744.
- [20] Bill R, Wirapati P, Messemaker M, et al. CXCL9:SPP1 macrophage polarity identifies a network of cellular programs that control human cancers. *Science*. 2023;381(6657):515-524.
- [21] Foley K, Rucki AA, Xiao Q, et al. Semaphorin 3D autocrine signaling mediates the metastatic role of annexin A2 in pancreatic cancer. *Sci Signal*. 2015;8(388):ra77.

- [22] Sheng W, Tang J, Cao R, Shi X, Ma Y, Dong M. Numb-PRRL promotes TGF- β 1- and EGF-induced epithelial-to-mesenchymal transition in pancreatic cancer. *Cell Death Dis.* 2022;13(2):173.
- [23] Hotz B, Arndt M, Dullat S, Bhargava S, Buhr HJ, Hotz HG. Epithelial to mesenchymal transition: expression of the regulators snail, slug, and twist in pancreatic cancer. *Clin Cancer Res.* 2007;13(16):4769-4776.
- [24] Sun L, Wang H, Wang Z, et al. Mixed lineage kinase domain-like protein mediates necrosis signaling downstream of RIP3 kinase. *Cell.* 2012;148(1-2):213-227.
- [25] Chauhan C, Martinez-Val A, Niedenthal R, et al. PRMT5-mediated regulatory arginine methylation of RIPK3. *Cell Death Discov.* 2023;9(1):14. Published 2023 Jan 19.
- [26] Zhan Q, Jeon J, Li Y, et al. CAMK2/CaMKII activates MLKL in short-term starvation to facilitate autophagic flux. *Autophagy.* 2022;18(4):726-744.

REVIEWER COMMENTS

Reviewer #2 (Necroptosis mediated immune regulation)(Remarks to the Author):

The authors provide additional data to show the effects of MLKL inhibition on PDAC growth, tumor migration and liver metastasis. Although they show photos of colony growth, migration and mets they do not quantify the tumor weight, tumor cellularity or number of mets. Its difficult to see that MLKL ko had any significant effects on metastasis. The other reagent used in the study is GW806742X, an MLKL and VEGFR2 inhibitor. Based on the role of VEGFR2, showing effects with MLKL ko is critical for these studies.

Have additional concerns about using CNV to identify cancer cells in mouse PDAC cells and how could one gene induces M1 polarization. Also unclear to me how MLKL O/E results in Caspase3/8 cleavage unless its just a read out of the cell death.

Reviewer #3 (Carcinoma)(Remarks to the Author):

no further comments.

REVIEWER COMMENTS

Reviewer #2 (Necroptosis mediated immune regulation)(Remarks to the Author):

The authors provide additional data to show the effects of MLKL inhibition on PDAC growth, tumor migration and liver metastasis. Although they show photos of colony growth, migration and mets they do not quantify the tumor weight, tumor cellularity or number of mets. Its difficult to see that MLKL ko had any significant effects on metastasis.

Response: Thank you for your thoughtful comments. We have added quantification plots and discussion to the revised figures as per your suggestion. (Fig S2a-S2f, Page 11 Line 7-12; Page 22 Line 16-18).

The other reagent used in the study is GW806742X, an MLKL and VEGFR2 inhibitor. Based on the role of VEGFR2, showing effects with MLKL ko is critical for these studies.

Response: Yes, GW806742X was also reported to inhibit VEGFR2^[1-2]. We agree its VEGFR2-inhibition role should be investigated to rule out confounding effects, though some prior reports utilizing GW806742X have failed to do it ^[3-10].

VEGFR2 is a crucial angiogenic factor. In this study, we examined the single-cell expression levels and found that it is predominantly expressed in endothelial cells. We observed no significant difference in VEGFR2 expression between the OE-Vector and OE-MLKL groups (Fig R1 below). To assess the VEGFR2 inhibitory effect of GW on orthotopic tumors, we conducted further analysis (Fig R2 below), and the result indicates that the VEGFR2-inhibitory role of GW is not significant at the doses administered. This may be because a single blockade of VEGFR2 may not be sufficient to completely inhibit angiogenesis. Just as existing TKIs, such as sorafenib and Lenvatinib, require the inhibition of multiple targets to exert sufficient angiogenesis inhibitory effects ^[11-12]. Furthermore, we observed that GW did not significantly affect the tumor burden of liver metastasis in the OE-Vector group mice (Fig2g), suggesting that GW did not inhibit VEGFR2 to inhibit liver metastasis in the OE-MLKL group.

We have observed a significant increase in angiogenesis in the OE-MLKL group due to the extensive activation of CXCL8, a well-known tumor angiogenesis-promoting cytokine ^[13] (Fig R2 below). However, it is interesting to note that the use of GW in the same dose has inhibited the increased angiogenesis in the OE-MLKL group (Fig R2 below). This result can be explained by the inhibitory effect of GW on necroptosis and subsequent CXCL8 activation. This finding further supports the potential application of GW in OE-MLKL PDAC.

Fig R1. The expression of VEGFR2 in scRNA-seq of C57BL orthotopic tumors. ns indicated that there is no statistically significant difference on VEGFR2 expression between the two groups.

CD34 IHC-staining in primary tumors of C57BL/6 orthotopic model

Fig R1. CD34 IHC-staining to reflect tumor angiogenesis in C57BL orthotopic model after different treatments (GW, 100 µM in 50 µl, iv. 3x/week). The mice were sacrificed 4 weeks after injection. n=8 per group.

We have carefully considered your suggestion of verifying the VEGFR2-inhibitory role of GW in KO-MLKL. However, we have determined that it may cause bias since KO-MLKL itself can cause complicated TME changes, making it difficult to determine whether the results are due to KO-MLKL or GW's VEGFR2-inhibitory role. As a result, we have decided to verify this in the OE-Vector group mice instead. We have discussed this matter in the revised manuscript. **(Page 21 Line 3-12; Page 25 Line1-6).**

Have additional concerns about using CNV to identify cancer cells in mouse PDAC cells and how could one gene induces M1 polarization.

Response: We apologize for the unclear description in our previous communication. We

conducted CNV analysis to identify cancer cells in orthotopic tumors. This was mainly due to two reasons: (1) when harvesting the samples for scRNA-seq analysis, it was inevitable to include some normal pancreas tissue; (2) we needed to accurately analyze the CD47 expression of cancer cells between the OE-Vector group and OE-MLKL group.

Differentiating tumor cells, stromal cells, and immune cells within TME is crucial in scRNA-seq ITH studies. CNV analysis is a classic method that identifies tumor cells by discerning their characteristic aneuploid copy numbers from the diploid copy numbers of normal cells. This enables the distinction of pancreatic cancer cells from normal ductal epithelial cells within the epithelial cell population, consistent with existing studies [14-16]. Thus, we used CNV analysis to classify epithelial cells into cancer cells and pancreatic ductal cells to minimize errors. We have updated the description in the revised manuscript (**Page 11, Line 22-26**).

We apologize for the lack of precision in the description of macrophages regarding M1 polarization. OE-MLKL increased the proportion of M1-macrophages, but not all macrophages were completely polarized into M1-macrophages. According to scRNA-seq, considerable M1-macrophages and M2-macrophages co-existed (**Fig3h**), which is consistent with Michael C Schmid et al [17]. We have provided a more appropriate expression in the revised manuscript (**Page 12, Line 5, 8-10, 17-18**).

Also unclear to me how MLKL O/E results in Caspase3/8 cleavage unless its just a read out of the cell death.

Response: We used Cleaved-Caspase 3 and Cleaved-Caspase 8 to determine the level of cell apoptosis between OE-Vector and OE-MLKL. The results showed a similar level of apoptosis [18-19]. Our intention was not to show that MLKL O/E leads to Caspase3/8 cleavage. We have provided a clearer description in the revised manuscript (**Page 9, Line 19-21**).

Reviewer #3 (Carcinoma)(Remarks to the Author):

no further comments.

Response: Thanks a lot.

Reference

- [1] Hildebrand JM, Tanzer MC, Lucet IS, et al. Activation of the pseudokinase MLKL unleashes the four-helix bundle domain to induce membrane localization and necroptotic cell death. *Proc Natl Acad Sci U S A*. 2014;111(42):15072-15077.
- [2] Sammond DM, Nailor KE, Veal JM, et al. Discovery of a novel and potent series of dianilinopyrimidineurea and urea isostere inhibitors of VEGFR2 tyrosine kinase. *Bioorg Med Chem Lett*. 2005;15(15):3519-3523.
- [3] Sun Y, Revach OY, Anderson S, et al. Targeting TBK1 to overcome resistance to cancer immunotherapy. *Nature*. 2023;615(7950):158-167.
- [4] Liu M, Lu J, Hu J, et al. Sodium sulfite triggered hepatic apoptosis, necroptosis, and pyroptosis by inducing mitochondrial damage in mice and AML-12 cells. *J Hazard Mater*. 2024;467:133719.
- [5] Zhan Q, Jeon J, Li Y, et al. CAMK2/CaMKII activates MLKL in short-term starvation to facilitate autophagic flux. *Autophagy*. 2022;18(4):726-744.
- [6] Zhong CS, Zeng B, Qiu JH, et al. Gout-associated monosodium urate crystal-induced necrosis is independent of NLRP3 activity but can be suppressed by combined inhibitors for multiple signaling pathways. *Acta Pharmacol Sin*. 2022;43(5):1324-1336.

- [7] Malireddi RKS, Sharma BR, Bynigeri RR, Wang Y, Lu J, Kanneganti TD. ZBP1 Drives IAV-Induced NLRP3 Inflammasome Activation and Lytic Cell Death, PANoptosis, Independent of the Necroptosis Executioner MLKL. *Viruses*. 2023;15(11):2141.
- [8] Deragon MA, McCaig WD, Truong PV, et al. Mitochondrial Trafficking of MLKL, Bak/Bax, and Drp1 Is Mediated by RIP1 and ROS which Leads to Decreased Mitochondrial Membrane Integrity during the Hyperglycemic Shift to Necroptosis. *Int J Mol Sci*. 2023;24(10):8609.
- [9] Dai X, Ma R, Jiang W, et al. Enterococcus faecalis-Induced Macrophage Necroptosis Promotes Refractory Apical Periodontitis. *Microbiol Spectr*. 2022;10(4):e0104522.
- [10] Wu W, Liu D, Zhao Y, et al. Cholecalciferol pretreatment ameliorates ischemia/reperfusion-induced acute kidney injury through inhibiting ROS production, NF- κ B pathway and pyroptosis. *Acta Histochem*. 2022;124(4):151875.
- [11] Liu LP, Ho RL, Chen GG, Lai PB. Sorafenib inhibits hypoxia-inducible factor-1 α synthesis: implications for antiangiogenic activity in hepatocellular carcinoma. *Clin Cancer Res*. 2012;18(20):5662-5671.
- [12] Yamamoto Y, Matsui J, Matsushima T, et al. Lenvatinib, an angiogenesis inhibitor targeting VEGFR/FGFR, shows broad antitumor activity in human tumor xenograft models associated with microvessel density and pericyte coverage. *Vasc Cell*. 2014;6:18.
- [13] Matsushima K, Yang D, Oppenheim JJ. Interleukin-8: An evolving chemokine. *Cytokine*. 2022;153:155828.
- [14] Wu SZ, Al-Eryani G, Roden DL, et al. A single-cell and spatially resolved atlas of human breast cancers. *Nat Genet*. 2021;53(9):1334-1347.
- [15] Izar B, Tirosh I, Stover EH, et al. A single-cell landscape of high-grade serous ovarian cancer. *Nat Med*. 2020;26(8):1271-1279.
- [16] Xue R, Zhang Q, Cao Q, et al. Liver tumour immune microenvironment subtypes and neutrophil heterogeneity. *Nature*. 2022;612(7938):141-147.
- [17] Astuti Y, Raymant M, Quaranta V, et al. Efferocytosis reprograms the tumor microenvironment to promote pancreatic cancer liver metastasis. *Nature cancer*, Feb 14, 2024. doi:10.1038/s43018-024-00731-2
- [18] Julien O, Wells JA. Caspases and their substrates. *Cell Death Differ*. 2017;24(8):1380-1389.
- [19] Marino-Merlo F, Klett A, Papaiani E, et al. Caspase-8 is required for HSV-1-induced apoptosis and promotes effective viral particle release via autophagy inhibition. *Cell Death Differ*. 2023;30(4):885-896.

REVIEWERS' COMMENTS

Reviewer #2 (Remarks to the Author):

The authors indirectly address the effects of GW on VEGFR signaling via their analysis of Vector only PDAC cells however, they provide no additional insight as to why MLKL is o/e in mouse and human PDAC lines. This remains a deficiency for this reviewer. Moreover several of the new SFigs are too small to view even when enlarged on screen (SFig 2k,g,h). And finally the ms will require editorial review. In several cases the wrong word is used in sentence and their meaning is lost.

REVIEWER COMMENTS

Reviewer #2 (Remarks to the Author):

The authors indirectly address the effects of GW on VEGFR signaling via their analysis of Vector only PDAC cells however, they provide no additional insight as to why MLKL is o/e in mouse and human PDAC lines. This remains a deficiency for this reviewer. Moreover several of the new SFIGs are too small to view even when enlarged on screen (SFIG 2k,g,h). And finally the ms will require editorial review. In several cases the wrong word is used in sentence and their meaning is lost.

Response: Thank you for your efforts and inputs in improving our manuscript. Regarding to the insights on why MLKL is o/e in T1M1-PDAC, in the first revision, we have investigated that as possible as we can based on the existed data, and we have indicated the potential reason as "aberrant splicing on MLKL pre-mRNA". It is quite interesting and we plan to study this further in our next research, but it is out of the main scope of this study.

In this resubmission, we have re-generated the figures and further polished the language according to your suggestion.